# A Probabilistic Framework For Solving High-Frequency Helmholtz Equation Via Diffusion Models

## Abstract

Deterministic neural operators perform well on many PDEs but can struggle with the approximation of high-frequency wave phenomena, where strong input-to-output sensitivity makes operator learning challenging and spectral bias blurs oscillations. We argue for a probabilistic approach, and use a conditional diffusion operator as a concrete tool to investigate it. Our study couples theory with practice: a theoretical sensitivity analysis explains why high frequency amplifies prediction errors, and suggests an evaluation protocol (including an energy-form metric) to test whether learned surrogates preserve *stable* quantities while capturing uncertainty. Across a range of regimes, the probabilistic neural operator is found to produce robust, full-domain predictions, better preserves energy at high frequency, and provides calibrated uncertainty that reflects input sensitivity, whereas deterministic approaches tend to oversmooth. These results position probabilistic operator learning as a principled and effective approach for solving complex PDEs such as Helmholtz in the challenging high-frequency regime.

## 1 Introduction

Helmholtz equation are a class of elliptic PDEs that arise in modeling time-harmonic wave propagation with a wide range of applications in applied silences from geophysics to medical fields such as imaging and therapeutic via ultrasound (Amundsen & Ursin, 2023; Huttunen et al., 2005; Sarvazyan et al., 2010; Rybyanets et al., 2015; Salahshoor et al., 2020; Juraev et al., 2024). Solving Helmholtz equation in high-frequency regimes in heterogeneous media require very fine computational grids that often renders the problem as computationally prohibitive (Chen, 2025; Bootland et al., 2021; Bao et al., 2004). This, in turn, motivates learning-based surrogates that remarkably reduce computational cost via fast inference while aiming to preserve fidelity. Among these, *operator learning* targets mappings between function spaces rather than pointwise inputs and outputs, with representative approaches including DeepONet and the Fourier Neural Operator (FNO), as well as physics-tailored extensions for wave physics and elasticity (Lu et al., 2021; Li et al., 2021; Zhang et al., 2023; Lehmann et al., 2024; Zou et al., 2024b; Chen et al., 2024; You et al., 2024). However, high-frequency wavefields in heterogeneous media expose two persistent limitations of *deterministic* operators: (i) *operator spectral bias*, wherein models preferentially capture low-frequency content and oversmooth oscillatory structure, degrading phase accuracy and interference patterns critical to acoustics and seismics (Fanaskov & Oseledets, 2023; Khodakarami et al., 2025; Xu et al., 2025); and (ii) *sensitivity* to small perturbations in inputs (e.g., sound speed, geometry, or frequency), which can induce multi-modality or sharply varying responses that point-estimate predictors neither represent nor calibrate (Ivanovs et al., 2021; Le & Dik, 2024; Behroozi et al., 2025). *Probabilistic* learning offers an alternative to *deterministic* (single-output) surrogates by modeling a distribution over solutions, thereby capturing multi-modality and input sensitivity (Stanziola et al., 2021; Vertes, 2020; Wu et al., 2022; Alkhalifah et al., 2021; Lobato et al., 2024). In this paper we use a conditional diffusion model as one concrete tool to examine probabilistic operator learning for high-frequency acoustics governed by the Helmholtz equation (Shysheya et al., 2024; Huang et al., 2024; Bastek et al., 2025b; Lim et al., 2023; Wang et al., 2024; Yao et al., 2025): we instantiate a *conditional diffusion operator* that maps problem inputs (sound-speed map, source mask, positional encodings) to the complex frequency-domain wavefield and evaluate it on a controlled 2D J-Wave benchmark spanning low to

high frequencies, comparing against strong deterministic baselines (FNO, HNO, and a backbone-matched U-Net). Our study revolves around two questions central to high-frequency prediction: (Q1) can a *probabilistic* operator preserve *stable* quantities—e.g., energies in sub-regions—more reliably than deterministic surrogates; and (Q2) can it produce *calibrated* uncertainty that faithfully reflects sensitivity to input perturbations? Although the underlying forward map is deterministic, practical indeterminacy at high frequencies makes single-output surrogates brittle. We therefore adopt a probabilistic formulation that learns a conditional distribution over solutions, using its mean for accuracy and its dispersion to encode epistemic sensitivity.

**Our contributions.** We use conditional diffusion models as probabilistic learning machines to: (i) learn fine-feature in solutions fields of PDE that are hard, if at all possible, to capture by any deterministic model, and (ii) to learn the push-forward of uncertainties in coefficient fields, reflected in our sensitivity study. Across all tested frequencies, diffusion achieves the lowest errors in $L^2$, $H^1$, and energy norm. We also demonstrate that diffusion model can predict the statistics of amplitudes in the far-field. Our sensitivity analysis further show that diffusion robustly mirrors the ground-truth variability and produces calibrated uncertainty, whereas deterministic operators are systematically under-dispersed and miss small-variance modes. Taken together, our results show that probabilistic operator learning is a promising approach for approximating high-frequency solutions of PDEs.

## 2 RELATED WORK

**Operator learning for Helmholtz.** Neural operators promise fast surrogates for PDEs at scale, but Helmholtz problems expose their core weakness: preserving high-frequency structure. Fourier Neural Operators can model elastic waves efficiently, yet their spectra skew low and tend to smooth oscillations—an instance of spectral bias (Zhang et al., 2023; Lehmann et al., 2024; Benítez et al., 2024). In response, Helmholtz-specific designs emerged. Helmholtz Neural Operators (HNO) move to the frequency domain, exchanging time stepping for per-frequency solves and delivering substantial memory/runtime benefits while retaining accuracy on elastic wavefields (Zou et al., 2024a;b). Complementary efforts build analytic structure into the surrogate: Neumann–series neural operators target large wavenumbers to improve stability and error (Chen et al., 2024), while multi-scale architectures inject resolution where spectral bias hurts most (You et al., 2024). Together, these works sharpen the central challenge—phase/interference fidelity under input sensitivity—and set the stage for probabilistic alternatives.

**Diffusion models for PDE solution fields.** Diffusion models approach the problem from the opposite direction: rather than a single point estimate, learn a *distribution* over solutions that can express multi-scale detail and uncertainty. Conditional diffusion has shown strong fidelity on challenging, high-frequency targets (Shysheya et al., 2024). The next question is *physics compliance*: physics-informed diffusion introduces residual penalties and priors in training/sampling (or via post-hoc distillation) so generated fields satisfy governing equations without sacrificing fidelity (Bastek et al., 2025b; Zhou et al., 2025; Zhang & Zou, 2025). When observations are partial, guided diffusion couples coefficients and solutions—enabling inference from sparse or indirect data and scaling across setups (Huang et al., 2024; Cao et al., 2025; Gao et al., 2024; 2025; Bergamin et al., 2025). Moving beyond grids, function-space diffusion defines priors directly on continuous fields, supporting discretization-agnostic learning and transfer across resolutions (Lim et al., 2023; Yao et al., 2025; Bastek et al., 2025a). Related score-based solvers extend these ideas to inverse problems and safe control with calibrated uncertainty (Haitsiukevich et al., 2024; Hu et al., 2025).

**Hybrid diffusion + operators for Helmholtz.** An innovation is to *condition* diffusion on a neural operator: the operator supplies coarse, global structure, while diffusion restores high-frequency detail and quantifies uncertainty. Such hybrids have reduced spectral bias in turbulence and fluid statistics (Oommen et al., 2025; Molinaro et al., 2024). For Helmholtz, physics-informed diffusion has been applied to radio-map generation with embedded Helmholtz constraints (Wang et al., 2025; Sortino et al., 2024; Jia et al., 2025), but these systems do not pair diffusion with a learned operator that maps coefficients to wavefields end-to-end. To our knowledge, a diffusion + operator hybrid *for Helmholtz operators* remains unexplored, so our work fills this gap by uniting operator learning with conditional diffusion to preserve phase-sensitive structure while providing calibrated uncertainty.

## 3 THEORETICAL FRAMEWORK

### 3.1 PROBABILISTIC OPERATOR LEARNING FOR PDES

Consider a boundary–value problem on an open, connected domain $\Omega \subset \mathbb{R}^d$ with boundary $\partial\Omega$,

$$\mathcal{L}_\kappa u = f \quad \text{in } \Omega, \qquad \mathcal{B} u = g \quad \text{on } \partial\Omega, \tag{1}$$

where $\mathcal{L}_\kappa$ is a differential operator depending on coefficient fields $\kappa$ (e.g., sound speed, density), $f$ is a source term, and $g$ encodes boundary data. Under standard well-posedness assumptions, equation 1 induces a *solution operator*

$$\mathcal{S} : \mathcal{Z} \to \mathcal{Y}, \qquad z := (\kappa, f, g) \in \mathcal{Z} \mapsto u = \mathcal{S}(z) \in \mathcal{Y}, \tag{2}$$

for suitable function spaces (e.g., $\mathcal{Z} \subset L^\infty \times L^2 \times H^{1/2}$, $\mathcal{Y} \subset H^1$). Classical solvers approximate $\mathcal{S}$ *per query* by discretizing equation 1 and computing $u_h \approx u$, which is expensive in many-query regimes (varying $z$ across geometries, coefficients, and frequencies). *Operator learning* seeks a data-driven surrogate $\widehat{\mathcal{S}}_\theta : \mathcal{Z} \to \mathcal{Y}$ trained from pairs $\{(z_i, u_i)\}_{i=1}^N$ with $u_i = \mathcal{S}(z_i)$, solving

$$\theta^\star \in \arg\min_\theta \frac{1}{N} \sum_{i=1}^N \mathcal{L}\big(\widehat{\mathcal{S}}_\theta(z_i), u_i\big),$$
$$\mathcal{L}(a, b) := \|a - b\|_\mathcal{Y}, \tag{3}$$

where $\| \cdot \|_\mathcal{Y}$ is the norm in the solution space. Popular *deterministic* architectures (e.g., Fourier Neural Operators (FNOs) (Li et al., 2021), DeepONets (Lu et al., 2021)) realize $\widehat{\mathcal{S}}_\theta$ via learned integral kernels or spectral multipliers and return a *single* prediction $\widehat{u}_\theta = \widehat{\mathcal{S}}_\theta(z)$ for input $z$. A *probabilistic* operator instead models the *conditional law* of the solution: For each input $z \in \mathcal{Z}$, define the conditional probability distribution $\mathbb{P}(u \mid z)$ of outputs $u$ given the conditioning variable $z$. The goal of probabilistic approaches, including conditional diffusion models, is to learn to (approximately) sample from this conditional probability distribution. In practice, we parameterize a generator $T_\vartheta$ and draw samples via Gaussian noise $\eta \sim \mathcal{N}(0, I)$,

$$u = T_\vartheta(\eta; z) \sim p_\vartheta(\cdot \mid z), \tag{4}$$

where the law $p_\vartheta(\cdot \mid z) = T_{\vartheta,\#}\mathcal{N}(0, I)$ is the push-forward of the reference Gaussian under the learned generator. After training $\vartheta$ with score-based objectives, inference then allows us to draw new samples from the learned conditional law. Typical probabilistic operators can be instantiated with conditional diffusion models, normalizing flows, or function-space priors (Lim et al., 2023; Wang et al., 2024; Yao et al., 2025; Shysheya et al., 2024; Huang et al., 2024; Chen & Vanden-Eijnden, 2025).

### 3.2 HIGH-FREQUENCY HELMHOLTZ AND WHY PROBABILISTIC HELPS

We consider the Helmholtz equation

$$c^2 \nabla^2 u + k^2 u = 0, \tag{5}$$

with inhomogeneous sound map $c = c(x)$. The Helmholtz equation underpins time-harmonic wave modeling in acoustics, elasticity, and scattering. Despite its ubiquity, its numerical solution is still facing many challenges. The reader is referred to Ernst & Gander (2011) for a detailed description of the challenges in solving Helmholtz equation. In a nutshell, Helmholtz operator is neither Hermitian symmetric nor coercive, and is poorly conditioned, which makes it hard to solve via iterative methods. With these difficulties in place, in the high wavenumber regime additionally requires resolving short wavelengths and controlling dispersion ("pollution") errors, which, collectively renders approximating high frequency solutions of Helmholtz a difficult problem. These difficulties have motivated studies on using neural operators for Helmholtz (Zhang et al., 2023; Lehmann et al., 2024; Benítez et al., 2024; Zou et al., 2024a;b).

Two obstacles for deterministic operators are aggravated by considering the high-frequency regime of the Helmholtz equations: (i) *spectral bias*, causing neural networks to preferentially fit low-frequency content and oversmooth oscillations, degrading phase/interference fidelity; and (ii) *input sensitivity*,

where tiny coefficient perturbations induce large phase shifts in the solution. Let $S : \mathcal{Z} \to \mathcal{Y}$ map a sound-speed field $c$ to the complex wavefield $u = S(c)$. Linearizing around a baseline $c_0$ yields

$$\delta u \approx DS[c_0](\delta c), \qquad L(c_0) := \|DS[c_0]\|_{\mathrm{op}}, \qquad \text{so} \quad \frac{\|\delta u\|}{\|u_0\|} \lesssim L(c_0)\frac{\|\delta c\|}{\|c_0\|}, \qquad (6)$$

with $u_0 := S(c_0)$. A 1D WKB argument, detailed in Appendix A.2, at large wavenumber $k$ makes the scaling explicit: along a ray parameterized by arclength $s \in [0, \ell]$ to $\boldsymbol{r}$, the WKB analysis implies that the wavefield $u$ and sound map $c$ are approximately related by

$$u(\boldsymbol{r}) \approx A_0 \exp\Big(\pm ik\int_0^\ell \frac{ds}{c(s)}\Big), \qquad (7)$$

so for small perturations $c = c_0 + \delta c$ with $\|\delta c\|_\infty / \|c_0\|_\infty \ll 1$, we obtain

$$\frac{\delta u(\boldsymbol{r})}{u_0(\boldsymbol{r})} \approx \exp\Big(\pm i\,\frac{k}{c_0^2}\int_0^\ell \delta c(s)\,ds\Big) - 1, \qquad \Big|\frac{\delta u(\boldsymbol{r})}{u_0(\boldsymbol{r})}\Big| \lesssim \underbrace{\frac{k\,\ell}{c_0}}_{=:L(k,\ell,c_0)}\frac{\|\delta c\|_\infty}{c_0}, \qquad (8)$$

(where $c_0$ is locally constant along the ray). Sensitivity thus grows linearly with frequency $k$ and propagation distance $\ell$ (and is minimal near the source). If the induced phase shift $\delta\phi(\boldsymbol{r}) \approx \frac{k}{c_0^2}\int_0^\ell \delta c(s)\,ds$ can be considered random (reflecting unresolved input variability/modeling error), the MSE-optimal *deterministic* predictor collapses oscillations by vector-averaging on the unit circle:

$$\hat{u}_{\mathrm{MSE}}(\boldsymbol{r}) = \mathbb{E}[u(\boldsymbol{r})\,|\,z] \approx u_0(\boldsymbol{r})\,\mathbb{E}\big[e^{i\delta\phi(\boldsymbol{r})}\big] \approx u_0(\boldsymbol{r})\,e^{-\frac{1}{2}\mathrm{Var}[\delta\phi(\boldsymbol{r})]}. \qquad (9)$$

The last exponential factor produces amplitude shrinkage, blurred interference, and lost high-frequency contrast—precisely when $L(k, \ell, c_0) \propto k\ell$ is large. This happens when either the wavenumber $k$ is large (high-frequency limit), or when the distance to the source $\ell$ is large.

A *probabilistic* operator instead models the conditional law $p_\vartheta(u\,|\,z)$ by internalizing phase ambiguity and input sensitivity rather than collapsing them into a single output. Drawing samples $\{u^{(s)}\}_{s=1}^S \sim p_\vartheta(\cdot\,|\,x)$ allows: (i) preserving oscillations and multi-modality at prediction time (via samples representing e.g. uncertain phase statistics, not just a learned mean); (ii) quantifying *calibrated uncertainty* that tracks sensitivity; and (iii) reporting *stable* functionals where phase cancels. For example, the energy

$$E(u) = \int_\Omega \Big(\|\nabla u(\boldsymbol{r})\|^2 + \frac{k^2}{c_0(\boldsymbol{r})^2}\,|u(\boldsymbol{r})|^2\Big)\,d\boldsymbol{r} \qquad (10)$$

can be estimated in a Bayes-optimal way for energy risk by $\hat{E} = \mathbb{E}[E(u)\,|\,z] \approx \frac{1}{S}\sum_{s=1}^S E(u^{(s)})$, which remains stable even when $\mathbb{E}[u\,|\,z]$ is attenuated by phase dispersion. Practically, $p_\vartheta(u\,|\,z)$ lets us decouple *fragile* quantities (phase, pointwise fields) from *stable* ones (energy, band-limited summaries), deliver coverage diagnostics for sensitivity-aware calibration, and choose decision rules matched to the evaluation metric—yielding robustness exactly where deterministic surrogates struggle.

## 3.3 CONDITIONAL DIFFUSION MODEL FOR PROBABILISTIC LEARNING

In this subsection, we describe our adopted conditional diffusion models for probabilistic learning of wavefields. Let us denote the Helmholtz solution as $u_0$, our objective is to approximate the conditional law $p_\theta(u_0\,|\,z)$ given the PDE inputs, where $u_0 \in \mathbb{C}^{H \times W}$ and $z = [\,c, m, \mathrm{PE}_x, \mathrm{PE}_y\,]$ concatenates the sound-speed map $c$, a binary source mask $m$, and sinusoidal positional encodings (we are assuming a 2D square domain for the sake of simplicity). This is achieved through *noising* and *denoising* process. For the first step, a *forward* noising chain progressively corrupts $u_0$ to $u_T$ as

$$q(u_t\,|\,u_{t-1}) = \mathcal{N}\big(u_t; \sqrt{\alpha_t}\,u_{t-1}, (1-\alpha_t)I\big), \quad q(u_t\,|\,u_0) = \mathcal{N}\big(u_t; \sqrt{\bar{\alpha}_t}\,u_0, (1-\bar{\alpha}_t)I\big), \qquad (11)$$

with schedule $\{\beta_t\}_{t=1}^T \subset (0,1)$, $\alpha_t = 1 - \beta_t$, and $\bar{\alpha}_t = \prod_{s \le t}\alpha_s$. Then, for the next step we use a time-indexed Gaussian family to approximate the *reverse* conditionals

$$p_\theta(u_{t-1}\,|\,u_t, z) = \mathcal{N}\big(u_{t-1}; \mu_\theta(u_t, t, z), \sigma_t^2 I\big),$$

$$\mu_\theta(u_t, t, z) = \frac{1}{\sqrt{\alpha_t}}\Big(u_t - \frac{\beta_t}{\sqrt{1-\bar{\alpha}_t}}\,\varepsilon_\theta(u_t, t, z)\Big), \qquad (12)$$

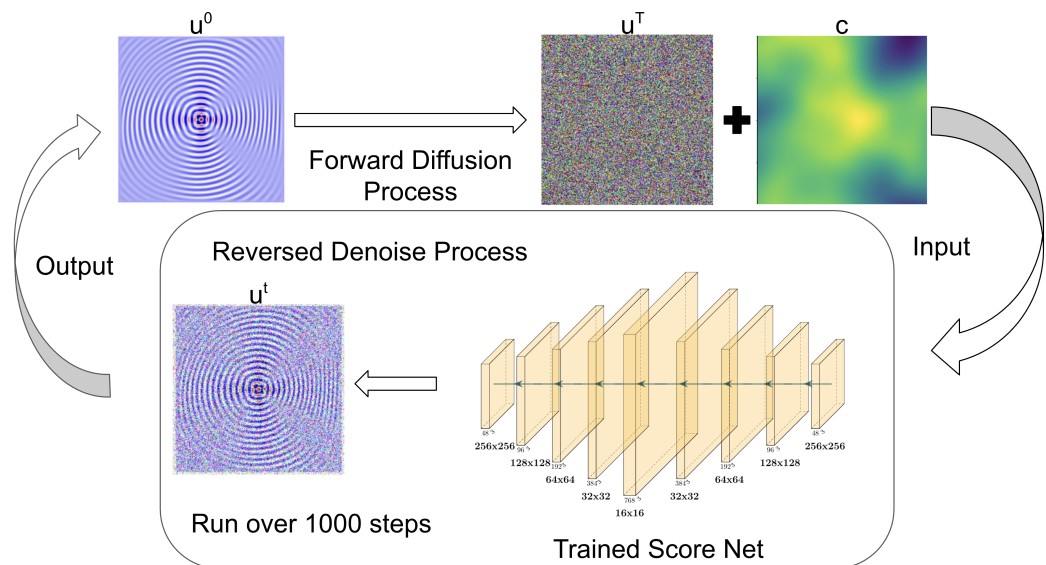

Figure 1: **Conditional diffusion for Helmholtz.** *Forward diffusion (top):* the wavefield $u_0$ is progressively noised by a fixed schedule to a Gaussian field $u_T$. *Reverse denoising (bottom):* sampling starts from pure Gaussian $u_T$ and is *conditioned* on the inputs $z$ comprised of sound-speed map $c$, as well as source mask and positional encodings that are not shown. A time-indexed U-Net ("Trained Score Net") predicts noise and removes it iteratively over $T \approx 1000$ steps to produce samples $u_0^{(s)}$ that approximate the conditional distribution of solutions.

where a U-Net with time embeddings parametrizes the noise predictor $\varepsilon_\theta$. We note that the conditioning $z$ is *never* noised. The learning objective aligns with conditional denoising score matching is:

$$\mathcal{L}(\theta) = \mathbb{E}_{u_0, z, t, \varepsilon} \left\| \varepsilon - \varepsilon_\theta\big(u_t, t, z\big) \right\|_2^2, \qquad u_t = \sqrt{\bar{\alpha}_t}\, u_0 + \sqrt{1 - \bar{\alpha}_t}\, \varepsilon, \ \ \varepsilon \sim \mathcal{N}(0, I), \qquad (13)$$

During the inference, we generate $S$ ancestral samples by iterating $u_T \sim \mathcal{N}(0, I)$ through $t = T \to 1$ to obtain $\{u_0^{(s)}\}_{s=1}^S \sim p_\theta(\cdot \mid z)$.

## 4 DETAILS OF DATASETS AND TRAINING

In this section we delineate the details of data generation and diffusion model training.

**Data generation.** For a fixed frequency, we synthesize a dataset comprised of 10000 pairs of sound speed maps and Helmholtz solution, i.e. $\{c_j(x), u_j(x)\}_{j=1}^{10000}$. Each $c_j$ is generated using a family of Gaussian random fields (GRFs; see Appendix B.1) where GRF parameters are randomly chosen from a range of amplitudes and correlation lengths. Out of the 10000 synthesized data, we allocate 8190, 1020, and 500 respectively for training, validation, and testing. The dataset is generated with recourse to the Helmholtz PDE solver *J-Wave*, which is a powerful spectral method based solver developed via JAX (Stanziola et al., 2022). In particular, for each sound speed map $c_j$, we solve the Helmholtz equation:

$$\left(\nabla^2 \ + \ \frac{\omega^2}{c_j(x)^2}\right)u_j \ = \ F, \qquad (14)$$

where $u_j$ is the complex pressure field at angular frequency $\omega = 2\pi f$ in a square domain with a $256 \times 256$ Cartesian grid, $F$ is a masked source term with a disk mask of radius $r_s = 10$ in terms of the grid points, and perfectly matched layers (PML) is implemented in the boundaries to eliminate the effects of boundary reflections. We repeat the aforementioned to generate six datasets for six different frequencies, ranging from $1.5 \times 10^5$ to $2.5 \times 10^6$ Hz. Without loss of generality, and to reflect a concrete application, both frequencies and sound speed ranges are chosen to reflect modeling ultrasonic waves in tissues. During the learning process, we expand the input to include normalized $c$, source mask, and sinusoidal positional encodings, with the wavefield solutions as target.

**Conditional diffusion for Helmholtz operator.** Unlike existing application of Diffusion models in time-dependent PDEs, such as PDE–DIFF (Shysheya et al., 2024), we learn a *solution operator* that corresponds to steady state waves, which means there is no temporal stacking and each training example is a single instance at a fixed frequency. We adopt the conditional denoising formulation, where only the target wavefield is noised, while the inputs are kept clean and given as conditioning. The backbone score net is a 2D U-Net with five down/up stages (widths $[48, 96, 192, 384, 768]$), three residual blocks per stage, LayerNorm, SiLU, and circular padding (Fig. 1). A sinusoidal time embedding is passed through an MLP ($128 \rightarrow 256 \rightarrow 64$) to produce a 64-D context vector. The conditioning tensor $C$ is used two ways: (i) concatenated to the input image stack, and (ii) mapped (via a linear layer) from the 64-D context to the current block's channel width to generate FiLM scale/shift $(\gamma, \beta) \in \mathbb{R}^f$, applied as $y = \gamma \odot h + \beta$. Lastly, the network predicts a single output channel (wavefield amplitude). See Figure 1 for an overview of the training/sampling pipeline and the FiLM-conditioned U-Net. For further details, the reader is referred to Appendix B).

Table 1: **Relative errors across frequencies.** Diffusion reports mean±std over $K=10$ samples.

| Frequency (Hz) | Metric | U-Net | FNO | HNO | Diffusion |
|---|---|---|---|---|---|
| 1.5e5 | $L^2$ | 0.040 | 0.167 | 0.115 | **0.027** ± 0.004 |
| | $H^1$ | 0.071 | 0.219 | 0.164 | **0.045** ± 0.004 |
| | Energy | 0.022 | 0.114 | 0.071 | **0.016** ± 0.003 |
| 2.5e5 | $L^2$ | 0.077 | 0.190 | 0.153 | **0.028** ± 0.001 |
| | $H^1$ | 0.121 | 0.278 | 0.219 | **0.044** ± 0.001 |
| | Energy | 0.026 | 0.086 | 0.075 | **0.013** ± 0.002 |
| 5e5 | $L^2$ | 0.083 | 0.176 | 0.133 | **0.018** ± 0.002 |
| | $H^1$ | 0.122 | 0.223 | 0.189 | **0.035** ± 0.002 |
| | Energy | 0.036 | 0.105 | 0.069 | **0.013** ± 0.005 |
| 1e6 | $L^2$ | 0.101 | 0.306 | 0.177 | **0.025** ± 0.005 |
| | $H^1$ | 0.150 | 0.352 | 0.254 | **0.040** ± 0.007 |
| | Energy | 0.033 | 0.091 | 0.082 | **0.017** ± 0.008 |
| 1.5e6 | $L^2$ | 0.464 | 0.363 | 0.398 | **0.046** ± 0.011 |
| | $H^1$ | 0.639 | 0.434 | 0.557 | **0.070** ± 0.015 |
| | Energy | 0.249 | 0.103 | 0.162 | **0.026** ± 0.015 |
| 2.5e6 | $L^2$ | 0.767 | 0.412 | 0.802 | **0.095** ± 0.019 |
| | $H^1$ | 1.000 | 0.494 | 0.996 | **0.135** ± 0.028 |
| | Energy | 0.514 | 0.141 | 0.590 | **0.036** ± 0.016 |

## 5 RESULTS

We present the results of our experiments under two categories: (i) error analysis: where we demonstrate capabilities of our probabilistic framework to outperform other neural operator techniques with quantitative results, and (ii) sensitivity analysis: where we systematically demonstrate that our approach enables capturing the perturbations incurred in wavefields via uncertainties in sound speeds.

### 5.1 ERROR ANALYSIS

We have investigated different diffusion frameworks with different noise scheduling, where we ablate diffusion samplers (computed *once* per setting) across step budgets $N_t \in \{10, 50, 100, 1000\}$: DDPM with both *linear* and *cosine* noise schedules, DDIM (cosine), and SDE (cosine) (in Appendix D). Among them, *DDPM with a cosine schedule* achieves the best accuracy at our highest step budget ($N_t = 1000$). We thus fix DDMP as the sampler of choice. To compare our results against deterministic frameworks, we have adopted Fourier Neural Operator (FNO) and the more recent Helmholtz Neural Operator (HNO), and trained them on the identical datasets used for the diffusion models (details of FNO and HNO can be found in Appendix B). We have computed and interrogated FNO, HNO, and backbone U-Net results as baseline deterministic approaches against our probabilistic framework. Table 7 summarizes the relative $L^2$ and $H^1$ errors for six different frequencies, ranging from low 150 kHz to 2.5 MHz. Since often in practice one is interested in energy, we have also quantified the relative energy errors defined as $\mathcal{E}_{\text{energy}}(\hat{u}, u) = |E(\hat{u}) - E(u)|/E(u)$ with $E(u) = \int_\Omega (\|\nabla u\|^2 + \frac{k^2}{c(x)^2}|u|^2) \, dx$. We note that diffusion results are obtained by drawing $K = 10$ samples

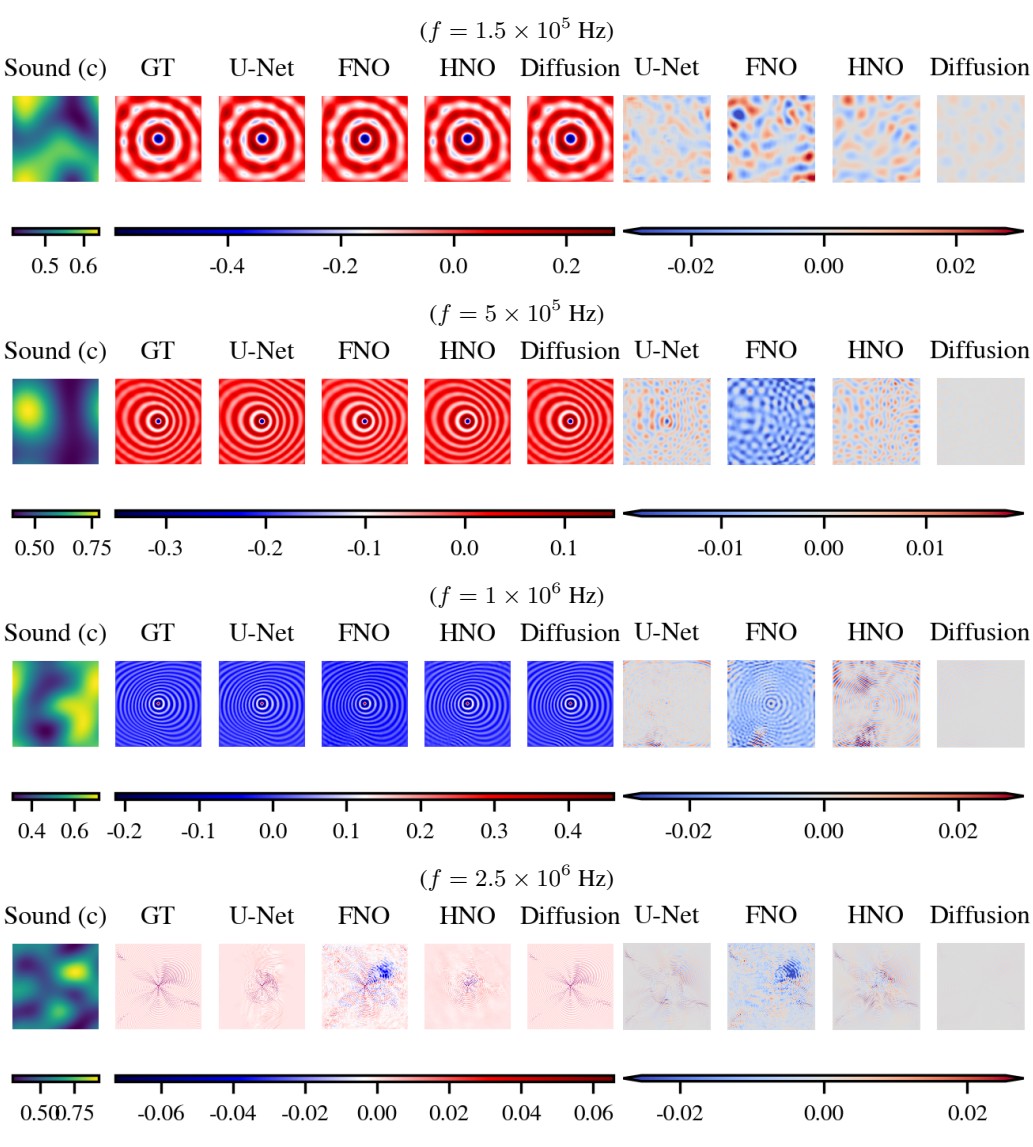

*Prediction vs. Ground Truth*                    *Residual (Pred−GT)*

Figure 2: **Qualitative comparisons across selected frequencies.** Each row shows, left-to-right, Sound Map (c), Ground Truth (GT), U-Net, FNO, HNO, Diffusion, followed by the comparison to GT.

per test input, and mean±std are reported. Our results demonstrate that diffusion models consistently yield smaller errors, and as frequency is ramped up, our probabilistic approach exhibits an order of magnitude lower errors compared to the best of three deterministic approaches. Notably, at 2.5 MHz, relative $L^2$ error for diffusion model is $0.095 \pm 0.019$, while the same quantity is computed for U-Net, FNO, and HNO as 0.767, 0.412, and 0.802, respectively. We further illustrate the computed wavefield solutions for a randomly chosen sound speed map. Figure. 2 shows the ground truth (GT), U-Net, FNO, HNO, and Diffusion results and their corresponding errors at a subset of four frequencies (see Appendix D.3 for more results). We observe that the Diffusion model outperforms the other deterministic approaches in capturing wave patterns and fine features of solutions.

## 5.2 SENSITIVITY ANALYSIS

To quantify how uncertainties in sound speed map is pushed forward to wavefields, we construct parametrized perturbation paths in coefficient function space as a homotopy from a randomly chosen

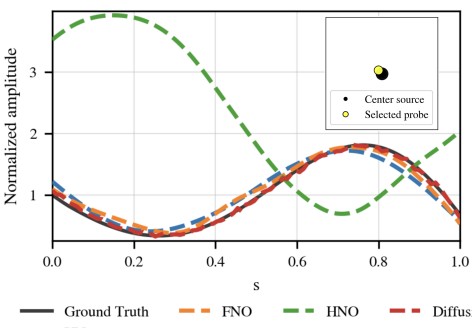 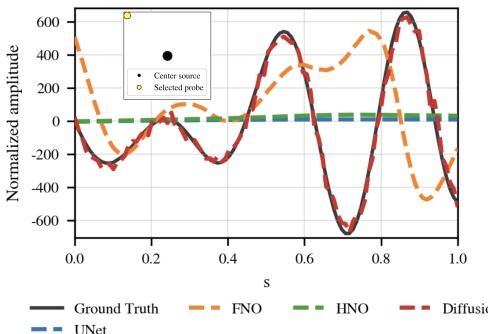

Figure 3: **Sampling along a parametrized path in coefficient function space.** We vary $s$ in $c_s = (1-s)c_0 + s\,c_1$ and track amplitude at fixed probe pixels (left: near source; right: near boundary).

reference medium to diverse realizations: starting from $c_0$ and $D{=}100$ GRF draws $\{c^{(d)}\}_{d=1}^D$, we define $c^{(d)}(s) = (1-s)c_0 + s\,c^{(d)}$ with $S{=}100$ steps $s \in [0,1]$ (as depicted in Figure 10 of the Appendix). For every $c^d(s)$, we then evaluate model's prediction $u_M(s,d;y,x)$ at 8 probe pixels (4 near the source center and 4 near the PML boundary). For the sake of brevity, we report one representative from each of the near and far locations in the main text (for all eight points we refer it in Appendix D). For small $s$, the slope along a path provides a local, directional view of sensitivity, since $\frac{d}{ds}u\big|_{s=0} \approx DS[c_0]\,[c^{(d)}-c_0]$, which ties directly to the bound $|\delta u/u_0| \lesssim (k\,r/c_0)\,(\|\delta c\|/c_0)$ (see section 3). As $s$ increases, the sound map becomes increasingly different from center $c_0$.

Placing probes both near and far allows a direct examination of the distance-to-source dependence $L \propto k\,r/c_0$. We utilize two complementary visualizations-sampling along a fixed direction $d$ to show rate-of-change along a single path, and kernel density estimate (KDE) (details in Appendix D.4) across directions at fixed $s$ to estimate the pushforward variability of $u$ under random coefficient perturbations. Hence, we interrogate sensitivity locally near $c_0$ (at $s{=}0$ and $s{=}0.1$) and at finite amplitudes (around $s \approx 1$) within one unified experimental design. At the highest frequency ($2.5{\times}10^6$ Hz), sampling plot reveals a clear separation: at the near-source probe, both diffusion model and majority of deterministic models can follow the ground-truth trajectory $u^\star(s,d^\star;y,x)$ across the entire path, matching both amplitude and phase; but at the near-boundary probe the discrepancy amplifies—only diffusion reproduces the rapid, non-monotone oscillations and turning points of the complex trajectory, while most deterministic models drift or flatten, consistent with the boundary oversmoothing in Figure 2 (Figure 3). In the *density plot* view, at $s{=}0$ all inputs satisfy $c^{(d)}(0) = c_0$, so the solver produces a single spike; deterministic models also produce a spike but it is *biased* relative to the solver's value, whereas diffusion yields a narrow yet non-degenerate predictive distribution that *covers* the ground-truth spike—reflecting the fact that at high frequency the map $c \mapsto u$ is effectively one-to-many (phase ambiguity, discretization/model mismatch, and measurement noise make several wavefields plausible for the same inputs). We turn this stochasticity to our benefit as diffusion model learns $p(u \mid x)$ rather than a single point, quantifying the push-forward distribution instead of committing to a potentially biased mean (Figure 4). At $s{=}1$, only diffusion captures the ground truth's broad amplitude distribution across directions, again consistent with the qualitative boundary behavior (Figures 2 and 4). We also note that for all sensitivity plots we use one diffusion sample per input to expose the predicted distribution rather than its mean.

## 6 DISCUSSION & CONCLUSION

We universally observe that diffusion models yield lower errors $L^2$ : at $2.5{\times}10^6$ Hz it preserves fine interference and overall energy while deterministic baselines (U-Net, FNO, HNO) oversmooth and lose phase coherence, especially at the boundary (Figure 2; Table 1). Two regularities emerge: errors increase from low to high frequency, and accuracy declines with distance from the source, consistent with the sensitivity lens $L \approx kr/c_0$ introduced earlier. By learning a conditional distribution over wavefields rather than a single point estimate, diffusion yields robust high-frequency predictions that better preserve the *stable* energy form and retain faint boundary patterns; sensitivity diagnostics

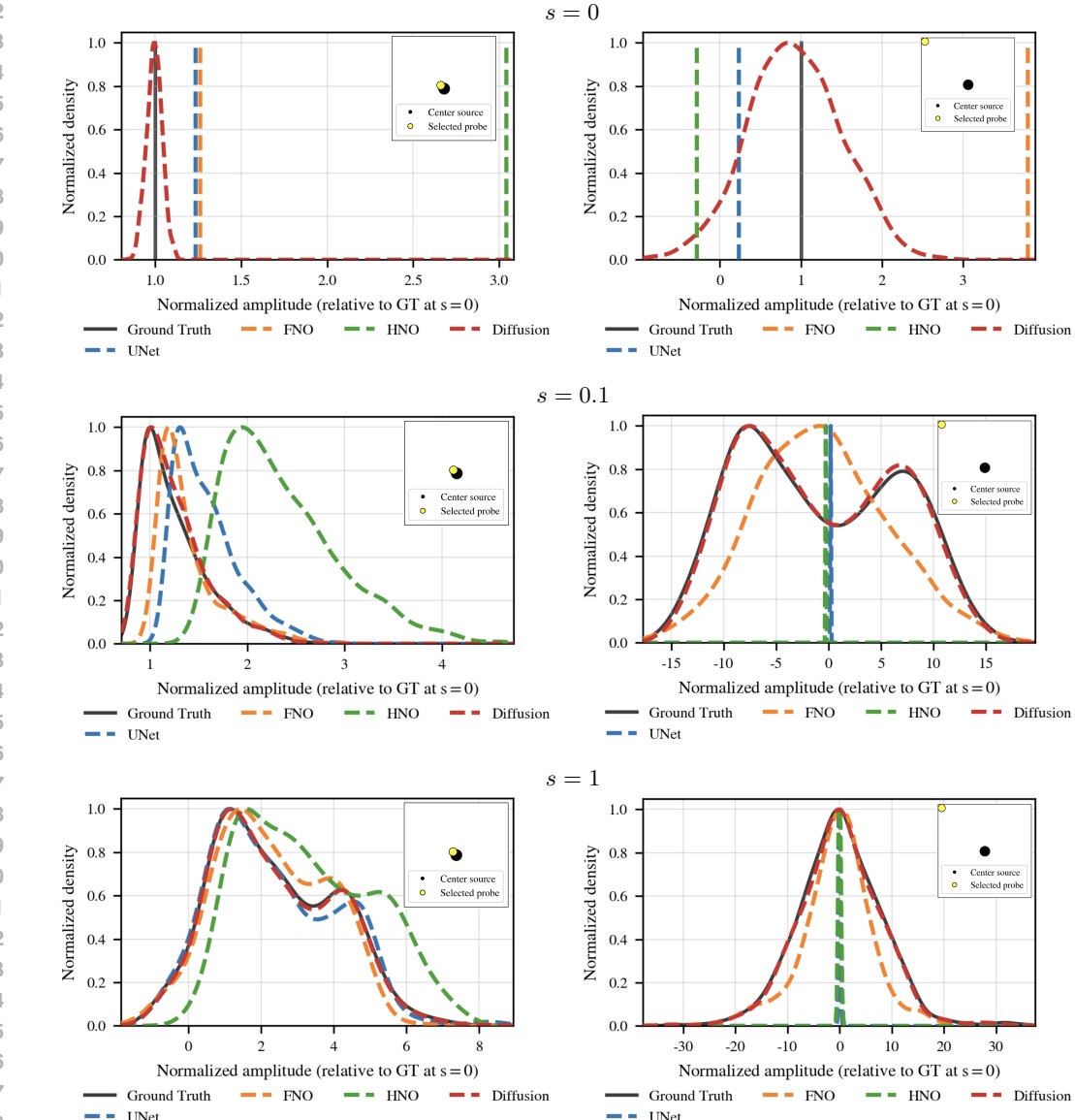

Figure 4: **Kernel density estimates across directions at selected interpolation levels.** Rows: $s = 0$, $s = 0.1$, $s = 1$. Left column: near the source. Right column: near the boundary.

(sampling along a fixed direction and density plots) corroborate these advantages by showing trajectory fidelity along perturbation paths and alignment with ground-truth amplitude distributions (Figures 3 and 4).

**Limitations & Future Work.** The main trade-off is sampling cost: our best configuration uses 1000 DDPM steps, which yields strong accuracy but slower inference. The current study is laying the groundwork and we envision accelerating the inference as a natural future work via flow matching training (Baldan et al., 2025), improved ODE/SDE solvers (Zhang & Chen, 2022), or latent/patchwise diffusion. Our study is currently restricted to a 2D domain with a single, centered point source and GRF-generated sound-speed fields; next, we will move beyond this controlled setting by (i) replacing GRFs with anatomically realistic, labeled brain media (including skull/soft-tissue heterogeneity and absorption) and (ii) testing multi-source/array configurations that better reflect therapeutic use. Finally, we will extend to full 3D Helmholtz propagation, addressing the associated memory and compute demands with multi-resolution architectures and efficient samplers, and we will reassess accuracy, energy-form stability, and uncertainty calibration under these more realistic conditions.

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

# A    DETAILED THEORY

## A.1    DATA GENERATION FOR 1D HELMHOLTZ

We complement the 2D study with a matched 1D Helmholtz experiment that uses the same data–model pipeline but replaces J-Wave for a finite-difference frequency-domain (FDFD) Helmholtz solver. Given samples of the wave speed, which is from the same GRF family as in 2D (varying length scales and amplitudes), we assemble the tridiagonal system for $-u'' + k(x)^2 u = 0$ with a left Dirichlet source and a right Sommerfeld (outgoing) boundary, and solve it in banded form (see Algorithm 1). Here we evaluate five frequencies $f \in \{1.5 \times 10^5, 2.5 \times 10^5, 5 \times 10^5, 7.5 \times 10^5, 1 \times 10^6\}$ Hz.

---

**Algorithm 1** 1D FDFD Helmholtz solver

---

**Require:** Wave speed samples $c[0{:}N{-}1]$ on $[0, L]$, frequency input (angular $\omega \leftarrow 2\pi f$)
**Ensure:** Complex field $u[0{:}N{-}1]$ solving $-u''(x) + k(x)^2 u(x) = 0$, $k(x){=}\omega/c(x)$
 1: $\Delta x \leftarrow L/(N-1)$
 2: **for** $j = 0$ to $N - 1$ **do**
 3:     $k[j] \leftarrow \omega/c[j]$
 4: **end for**
 5: Allocate `ab` $\in \mathbb{C}^{3 \times N}$ and $b \in \mathbb{C}^N$; initialize to 0
 6: Interior stencil (implicit via diagonals):
 7:     Set `ab`$[0, j{-}1] \leftarrow 1$, `ab`$[1, j] \leftarrow -2 + (k[j]\Delta x)^2$, `ab`$[2, j{+}1] \leftarrow 1$ for $j = 1, \ldots, N-2$
 8: **Left BC (Dirichlet)** $u(0){=}\omega$:
 9:     `ab`$[1, 0] \leftarrow 1$; `ab`$[0, 1] \leftarrow 0$; $b[0] \leftarrow \omega$
10: **Right BC (Sommerfeld)** $u'(L) + i\,k(L)u(L){=}0$:
11:     `ab`$[1, N{-}1] \leftarrow i\,k[N{-}1]\Delta x - 1$; `ab`$[2, N{-}1] \leftarrow 1$; $b[N{-}1] \leftarrow 0$
12: Solve `ab` $\cdot u = b$ with a tridiagonal banded solver (e.g., `solve_banded((1,1), ab, b)`)
13: **return** $u$

---

## A.2    1D WKB ANALYSIS

We consider the 1D variable-coefficient Helmholtz problem on $x \in [0, \ell]$,

$$u''(x) + k^2 n(x)^2\, u(x) \;=\; 0, \qquad n(x) := \frac{1}{c(x)}, \quad k := \frac{\omega}{\bar{c}}, \tag{15}$$

with a choice of boundary conditions that select a left-going branch. In the high-frequency regime ($k \gg 1$) and under the standard WKB assumptions (smooth $c$, no turning points, a single ray without caustics), we seek a solution of the form

$$u(x) \;=\; a(x)\, \exp\!\big(ik\,\phi(x)\big). \tag{16}$$

Substituting equation 16 into equation 15 and balancing powers of $k$ yields the eikonal and transport equations:

$$\mathcal{O}(k^2): \quad \big(\phi'(x)\big)^2 \;=\; n(x)^2, \qquad \Rightarrow \quad \phi'(x) = \pm n(x), \tag{17}$$

$$\mathcal{O}(k): \quad 2a'(x)\,\phi'(x) + a(x)\,\phi''(x) \;=\; 0 \;\Rightarrow\; a(x)\,\sqrt{\phi'(x)} = \text{const.} \tag{18}$$

Defining the *travel time* (phase)

$$\tau(x) \;:=\; \int_0^x n(s)\,ds \;=\; \int_0^x \frac{ds}{c(s)}, \tag{19}$$

and absorbing the slow amplitude factor into a constant $A_0$ (this is justified since $a'(x) = \mathcal{O}(1)$ while phase varies at rate $k$), the leading-order WKB solution reads

$$u(x) \;\approx\; A_0 \exp\!\big(\pm ik\,\tau(x)\big). \tag{20}$$

Then, perturbate the phase by small coefficient changes: let $c_0$ be a baseline field with $u_0(x) \approx A_0 e^{\pm ik\tau_0(x)}$ where $\tau_0(x) = \int_0^x ds/c_0(s)$. For a small perturbation $c = c_0 + \delta c$ with

$\|\delta c\|_\infty / \|c_0\|_\infty \ll 1$, the refractive index $n = 1/c$ satisfies

$$n(x) \;=\; \frac{1}{c_0(x) + \delta c(x)} \;=\; n_0(x) + \delta n(x), \qquad n_0(x) = \frac{1}{c_0(x)}, \qquad \delta n(x) \;=\; -\frac{\delta c(x)}{c_0(x)^2} + \mathcal{O}(\delta c^2). \tag{21}$$

Hence the perturbed travel time $\tau(x) = \tau_0(x) + \delta\tau(x)$ obeys

$$\delta\tau(x) \;=\; \int_0^x \delta n(s)\, ds \;=\; -\int_0^x \frac{\delta c(s)}{c_0(s)^2}\, ds \;+\; \mathcal{O}(\delta c^2). \tag{22}$$

To leading order (neglecting the subleading amplitude change), the perturbed field is

$$u(x) \;\approx\; A_0 \exp\Big( \pm ik\big[\tau_0(x) + \delta\tau(x)\big]\Big) \;=\; u_0(x)\, \exp\big( \pm ik\, \delta\tau(x)\big). \tag{23}$$

Thus the *relative* perturbation is phase-dominated:

$$\frac{\delta u(x)}{u_0(x)} \;=\; \exp\big( \pm ik\, \delta\tau(x)\big) - 1 \;=\; \pm ik\, \delta\tau(x) + \mathcal{O}\big((k\, \delta\tau)^2\big). \tag{24}$$

Bounding equation 22 by $\|\delta c\|_\infty$ and $c_{0,\min} := \inf_{s \in [0,\ell]} c_0(s)$ gives

$$|\delta\tau(x)| \;\leq\; \int_0^x \frac{|\delta c(s)|}{c_0(s)^2}\, ds \;\leq\; \frac{x}{c_{0,\min}^2}\, \|\delta c\|_\infty \;\leq\; \frac{\ell}{c_{0,\min}^2}\, \|\delta c\|_\infty. \tag{25}$$

Combining with equation 24 yields the high-frequency sensitivity estimate

$$\left| \frac{\delta u(x)}{u_0(x)} \right| \;\lesssim\; k\, \frac{\ell}{c_{0,\min}^2}\, \|\delta c\|_\infty \qquad \text{(to leading order in } \|\delta c\|_\infty\text{)}. \tag{26}$$

In the special case of *constant* baseline speed $c_0$, equation 26 simplifies to

$$\left| \frac{\delta u(x)}{u_0(x)} \right| \;\lesssim\; \underbrace{\frac{k\, \ell}{c_0}}_{=:L(k,\ell,c_0)}\, \frac{\|\delta c\|_\infty}{c_0}, \tag{27}$$

which makes explicit that the effective local Lipschitz factor grows linearly with wavenumber and path length.

As connection to the previous discussion of linearized operator sensitivity, recall the linearization of the solution map $S : \mathcal{Z} \to \mathcal{Y}$ at $c_0$,

$$\delta u \;\approx\; DS[c_0]\, (\delta c), \qquad L(c_0) := \|DS[c_0]\|_{\mathrm{op}}, \qquad \frac{\|\delta u\|}{\|u_0\|} \;\lesssim\; L(c_0)\, \frac{\|\delta c\|}{\|c_0\|}. \tag{28}$$

The WKB bound equation 27 shows that, along a single ray of length $\ell$, the *phase* contribution enforces

$$L(c_0) \;\gtrsim\; \frac{k\, \ell}{c_0} \quad \text{(constant } c_0\text{)}, \qquad \text{or more generally} \qquad L(c_0) \;\gtrsim\; k\, \ell\, c_{0,\min}^{-2}\, \|c_0\|_\infty. \tag{29}$$

Consequently, even tiny relative coefficient errors produce order-one relative wavefield errors when $k\ell \gg 1$ (large phase accumulation). Deterministic $\ell_2$-trained operators therefore average over phase-inconsistent targets and attenuate oscillations, whereas probabilistic models that learn the conditional law $p(u \,|\, c)$ can sample a coherent phase mode consistent with $c$, preserving high-frequency interference.

# B  METHOD DETAILS

## B.1  GAUSSIAN RANDOM FIELDS

We synthesize heterogeneous acoustic media by sampling stationary Gaussian random fields (GRFs) on the simulation grid and mapping them to sound speed. Specifically, we construct a mean-zero GRF in the Fourier domain by filtering complex white noise with an exponential spectral envelope and then applying an inverse real FFT (see Algorithm 2). This yields families of coefficient fields with controllable smoothness and correlation length, providing ground-truth dataset across frequencies for the Helmholtz operator-learning benchmarks in the main text.

---

**Algorithm 2** Sampling heterogeneous sound-speed fields via spectral GRFs

---

**Require:** Grid size $(N_x, N_y)$; background speed $c_{\text{bg}}$; scale $\sigma_c$; bounds $[c_{\min}, c_{\max}]$

1: Sample hyperparameters: $\alpha \sim \mathcal{U}(0.5, 2.5), \; \ell \sim \mathcal{U}(0.35, 0.7)$

2: Construct frequency grids $k_x = \text{fftfreq}(N_x), k_y = \text{rfftfreq}(N_y)$; form mesh $K = \sqrt{k_x^2 + k_y^2}$

3: Define spectral envelope $\lambda \leftarrow \exp\big(-(\ell K)^\alpha\big)$

4: **repeat**

5:     Draw complex white noise $\eta \leftarrow \eta_{\text{r}} + i\,\eta_{\text{i}}$, with $\eta_{\text{r}}, \eta_{\text{i}} \overset{\text{i.i.d.}}{\sim} \mathcal{N}(0,1)$

6:     Spectral field $\hat{u} \leftarrow \lambda \odot \eta$

7:     Realization $u \leftarrow \text{irfft2}(\hat{u}; N_x, N_y)$;   mean–center: $u \leftarrow u - \text{mean}(u)$

8:     Map to sound speed: $c(x) \leftarrow c_{\text{bg}} + \sigma_c\, u(x)$

9: **until** $c_{\min} < c(x) < c_{\max}$ for all grid points $x$

10: **return** $c$

---

B.2    DENOISER ARCHITECTURE (CONDITIONAL U-NET)

**Convolution (Conv1d/Conv2d).** Let $z \in \mathbb{R}^{H \times W \times C}$ be an input tensor and $K \in \mathbb{R}^{k \times k \times C \times \widehat{C}}$ a kernel bank producing $\widehat{C}$ output channels. For stride $s \in \mathbb{N}$, the discrete multichannel convolution $\mathcal{C} : \mathbb{R}^{H \times W \times C} \to \mathbb{R}^{\widehat{H} \times \widehat{W} \times \widehat{C}}$ is

$$\big(\mathcal{C}(z)\big)[i, j, \widehat{\ell}] = \sum_{m=0}^{k-1} \sum_{n=0}^{k-1} \sum_{\ell=1}^{C} K[m, n, \ell, \widehat{\ell}]\, z[i\,s + m,\; j\,s + n,\; \ell], \quad \begin{matrix} i = 0, \dots, \widehat{H}-1, \\ j = 0, \dots, \widehat{W}-1, \\ \widehat{\ell} = 1, \dots, \widehat{C}. \end{matrix} \quad (30)$$

(To handle outofbounds indices, we use *circular* padding to mitigate boundary artifacts in phaselike Helmholtz fields.) The encoder downsamples with $k=3$, stride $s=2$ (no stride at the first scale), while the decoder mirrors this via nearestneighbor upsampling by a factor 2 followed by a $k=3$ refinement convolution.

**Activation functions (SiLU).** We use the SiLU nonlinearity with $\beta=1$. For a scalar $z$,

$$\text{SiLU}(z) = z\,\sigma(z) = \frac{z}{1 + e^{-z}},$$

where $\sigma(z)$ is the logistic sigmoid. SiLU is smooth and *self-gating* (the input modulates its own pass-through), and unlike ReLU it preserves small negative activations.

**Normalization.** To stabilize training with small batches typical in PDE settings, we normalize activations either by *group normalization* (GN) or *layer normalization* (LN).

*Group Normalization (GN).* Given $z \in \mathbb{R}^{H \times W \times C}$, split the channel axis into $G$ groups of size $C/G$. Let $\mathcal{G}_g$ be the index set of group $g$ with $m = HW \cdot (C/G)$. Per–group statistics and normalization are

$$\mu_g = \frac{1}{m} \sum_{(h,w,c) \in \mathcal{G}_g} z_{h,w,c}, \qquad \sigma_g^2 = \frac{1}{m} \sum_{(h,w,c) \in \mathcal{G}_g} (z_{h,w,c} - \mu_g)^2, \quad (31)$$

$$\widehat{z}_{h,w,c} = \frac{z_{h,w,c} - \mu_g}{\sqrt{\sigma_g^2 + \varepsilon}}, \quad (h,w,c) \in \mathcal{G}_g, \qquad \widetilde{z}_{h,w,c} = \gamma_c\, \widehat{z}_{h,w,c} + \beta_c, \quad (32)$$

with learnable per–channel scale $\gamma_c$ and shift $\beta_c$.

*Layer Normalization (LN).* LN normalizes *all* features of a sample jointly (channel+space). Let $\mathcal{I} = \{(h,w,c) : 1 \le h \le H, 1 \le w \le W, 1 \le c \le C\}$ and $M = HWC$. Then

$$\mu = \frac{1}{M} \sum_{(h,w,c) \in \mathcal{I}} z_{h,w,c}, \qquad \sigma^2 = \frac{1}{M} \sum_{(h,w,c) \in \mathcal{I}} (z_{h,w,c} - \mu)^2, \quad (33)$$

$$\widehat{z}_{h,w,c} = \frac{z_{h,w,c} - \mu}{\sqrt{\sigma^2 + \varepsilon}}, \qquad \widetilde{z}_{h,w,c} = \gamma_c\, \widehat{z}_{h,w,c} + \beta_c. \quad (34)$$

For simplicity, in our code we use LayerNorm by default.

**Diffusion time embedding.** The denoiser conditions on diffusion time $t \in [0, 1]$ via a sinusoidal Fourier feature map followed by a small MLP:

$$\phi(t) = \left[ \cos(\omega_1 t), \ldots, \cos(\omega_{64} t),\ \sin(\omega_1 t), \ldots, \sin(\omega_{64} t) \right] \in \mathbb{R}^{128}, \quad \omega_r = \tfrac{\pi}{2} \cdot 10^{3 \frac{r-1}{63}}. \quad (35)$$

This produces multi–scale time features spanning several orders of magnitude. Another MLP then maps $\phi(t)$ to the per–block conditioning width:

$$e(t) = \text{Linear}_{128 \to 256}\big(\phi(t)\big) \xrightarrow{\text{SiLU}} \text{Linear}_{256 \to d} \in \mathbb{R}^d, \quad (36)$$

after which a per–scale linear projection broadcasts $e(t)$ over space and injects it additively into each *context residual block* (FiLM–style conditioning) prior to the two $k{=}3$ convolutions. This ensures that denoising decisions are time–aware at every resolution.

**Positional encodings for coordinates.** In addition to time conditioning, we concatenate sinusoidal *spatial* encodings to the input channels (part of $z$): in 2D, $\{\sin(2^\ell \pi x), \cos(2^\ell \pi x), \sin(2^\ell \pi y), \cos(2^\ell \pi y)\}_{\ell=0}^{L-1}$. These multi–frequency positional cues help align phase and interference patterns across the domain, particularly at high Helmholtz wavenumber $k$.

**U -Net layout.** Here we specify implementation choices that were only summarized in the paper. The backbone is an encoder–decoder with skip connections and *context residual blocks* that inject the diffusion time embedding $e(t)$ at every depth. At scale $s$ with channel width $C_s$, a block applies FiLM–style additive conditioning and two same–width convolutions with normalization and SiLU:

$$z \leftarrow z + \underbrace{\Big(\text{Conv}_3\big(\sigma(\text{Norm}(z + \Pi_s e(t)))\big) \to \text{Conv}_3\Big)}_{\text{residual branch}},$$

where $\Pi_s : \mathbb{R}^d \to \mathbb{R}^{C_s}$ is a linear projection of $e(t)$ followed by unflattening broadcast over space, $\text{Conv}_3$ denotes a kernel-3 circularly padded convolution (Conv1d/2d depending on dimensionality), Norm is LayerNorm by default, and $\sigma$ is SiLU. Downsampling "heads" are strided convolutions (kernel 3, stride 2 at all but the first scale); upsampling "tails" perform $\text{Norm} \to$ nearest–neighbor upsample $\times 2 \to$ kernel-3 convolution and are added to the aligned encoder feature (*skip connection*). Circular padding is used throughout to reduce boundary artifacts for Helmholtz phase fields.

**Training** We train the conditional U -Net as a score network within a variance–preserving SDE (VPSDE) using mini-batch denoising score matching on tensors $X \in \mathbb{R}^{B \times C \times H \times W}$ with $H = W = 256$ and a small batch size $B = 32$; the channel stack is a *single* solution channel that is noised, and *three* clean conditioning channels (e.g., sound speed, source mask, positional encodings). At each iteration we sample $t \sim \text{Unif}(0, 1)$, obtain $u_t$ via the VPSDE perturbation kernel implemented in our forward function, and minimize a mean-squared denoising objective,

$$\mathcal{L}_{\text{VPSDE}}(\theta) = \mathbb{E}_{u_0,\, z,\, t}\Big[ w(t) \big\| s_\theta\big(u_t, z, t\big) - \tilde{y}(u_t, t) \big\|_2^2 \Big], \quad (37)$$

where $s_\theta$ is the U -Net denoiser, $\tilde{y}(u_t, t)$ is the VPSDE target produced by forward diffusion, and $w(t) \equiv 1$ in our runs; in code this appears as $\texttt{mse} = (\texttt{net}(u_t, z, t) - \texttt{output})^2$ averaged over spatial/channel dimensions, followed by $\texttt{mean()}$ over the batch. Each epoch iterates over mini-batches on the selected GPU device, updates an EMA of parameters for more stable sampling with a standard learning-rate scheduler, and evaluates the same loss on a held-out validation split without gradients. At inference, we use different sampling methods (see more details in Appendix D.2).

### B.3 Fourier Neural Operator (FNO)

We implement a 2D *tensorized* Fourier neural operator as a strong deterministic baseline. Let $z \in \mathcal{Z} \subset \mathbb{R}^{H \times W \times d_u}$ denote the input stack of channels (here $d_z{=}3$: sound speed, source mask, positional encodings) and $G_\phi : \mathcal{Z} \to \mathcal{Y}$ the learned operator producing the target wavefield. The FNO is composed as

$$G_\phi = Q \circ L_L \circ \cdots \circ L_1 \circ R,$$

with a *lifting* $R : \mathbb{R}^{d_u} \to \mathbb{R}^{d_v}$ (pointwise $1{\times}1$ conv) that increases channel width to $d_v$, followed by $L$ Fourier layers $L_\ell$, and a *projection* $Q : \mathbb{R}^{d_v} \to \mathbb{R}^{d_y}$ (pointwise head) back to output channels. Each Fourier layer is

$$v_{\ell+1}(z) = \sigma\Big(W_\ell v_\ell(z) + \mathcal{F}^{-1}\big[ P_\ell(k) \cdot \mathcal{F}[v_\ell](k) \big](z)\Big), \quad (38)$$

where $v_\ell \in \mathbb{R}^{H \times W \times d_v}$, $W_\ell$ is a learned *local* (linear/skip) operator, $\mathcal{F}$ is the 2D FFT, and $P_\ell(k) \in \mathbb{C}^{d_v \times d_v}$ are learned complex multipliers applied only on a truncated band of modes. Here we retain $(n_{\text{modes}}^h, n_{\text{modes}}^w) = (64, 64)$ in every Fourier layer and use GeLU nonlinearity (`preactivation=0`); the spectral update is the factorized/tensorized variant that parameterizes $P_\ell(k)$ with low-rank factors (`rank=1.0`), reducing memory and improving stability at $256 \times 256$. After each spectral block, a lightweight channel MLP (`use_channel_mlp=1`, expansion=0.5, dropout=0) mixes features in the spatial domain. We employ group normalization and linear residual skips, while domain padding is off by default. The architecture uses $L = 4$ Fourier layers with hidden width $d_v = 32$ and projection ratio $2\times$ inside the spectral block, taking `data_channels=3` as input and producing a single output channel per component. Optimization follows standard operator-learning practice: 1000 epochs with AdamW (lr $= 5 \times 10^{-3}$, weight decay $= 10^{-4}$), batch size 32, `StepLR` scheduler (step size=60, $\gamma = 0.5$), and an $H^1$ training loss to encourage gradient fidelity. Overall, this `FNO` baseline provides a fair deterministic comparator focused on frequency-domain accuracy under controlled memory/compute.

### B.4 HELMHOLTZ NEURAL OPERATOR (HNO)

Our HNO baseline follows a UNO–style spectral operator with U–shaped skip connections tailored to $256 \times 256$ grids. Inputs are tensors $z \in \mathbb{R}^{B \times H \times W \times C}$ with $H = W = 256$ and $C = 3$ channels (sound speed, source mask, positional features); outputs are $u \in \mathbb{R}^{B \times H \times W \times 1}$. Then, we concatenate a coordinate grid (linear $[0, 1]$) to the input, lift with two pointwise MLPs (`fc_n1`, `fc0`), permute to $(B, C, H, W)$, and pass through eight *operator blocks* with encoder–decoder topology and skip concatenations. Each block is

$$\text{OperatorBlock2D:} \quad \underbrace{\text{SpectralConv2d\_Uno}}_{\text{FFT} \to \text{low-rank spectral map} \to \text{iFFT}} + \underbrace{\text{pointwise\_op\_2D}}_{\text{1X1 conv + bicubic resize}}$$

$$\xrightarrow{\text{GELU}} \text{MLP(1x1)} \xrightarrow{\text{LayerNorm}} \text{residual add} \xrightarrow{\text{GELU}} .$$

The spectral layer performs $z \mapsto \mathcal{F}^{-1}\big(P(\mathbf{k}) \odot \mathcal{F}z\big)$ with two learnable complex weight tensors (`weights1`, `weights2`) applied to the positive and negative vertical bands of retained modes. In parallel, a pointwise 1x1 convolution is up-/down–scaled via bicubic interpolation to the block's output resolution and added to the spectral path, improving locality and stabilizing high–$k$ content. Each block ends with an MLP (two 1x1 convs with GeLU) and `LayerNorm` over $[H, W]$, followed by a residual addition with the pointwise branch and GeLU. The U–shaped pathway uses resolutions $(256, 256) \to (128, 128) \to (64, 64) \to (32, 32)$ and back, with mode budgets matched per scale: `conv1`: $(\dim = 256, 256; \text{modes} = 192, 96)$, `conv2`: $(128, 128; 128, 64)$, `conv3`: $(64, 64; 64, 32)$, `conv4`: $(32, 32; 32, 16)$, then symmetric values on the way up. After the decoder, features are concatenated with the lifted input, projected by a kernel MLP to output a per–pixel scalar kernel, and contracted with features via a normalized Einstein sum, yielding a per–pixel reduction. A final two–layer head with `tanh` maps to outputs.

During the training, we instantiate `UNO2D` with `width=16`, `in_channels=3`, `out_channels=1` and train on $256 \times 256$ fields using Adam (lr $10^{-3}$, weight decay $10^{-5}$), `StepLR` (step=30, $\gamma$=0.5), batch size 32, for up to 1000 epochs. The loss combines $0.9\,L^1 + 0.1\,L^2$ on the real component trained per pass. This HNO baseline thus realizes a frequency–domain, multi–scale spectral operator that couples learned Fourier multipliers with local pointwise updates and U–shaped skips, providing a strong deterministic comparator for our probabilistic diffusion operator on Helmholtz problems.

## C PRELIMINARY EXTENTISON TO 3D HELMHOLTZ

### C.1 NETWORK BACKBONE – UViT

Our UViT backbone, illustrated in Fig. 5, is a U-shaped vision transformer inspired by Molinaro et al. (2024) and adapted here for Helmholtz wavefields. The network operates on multi-scale feature maps along an encoder–decoder hierarchy, and each resolution level applies a stack of residual blocks (blue), optionally followed by self-attention blocks (red) at selected scales. Downsampling and

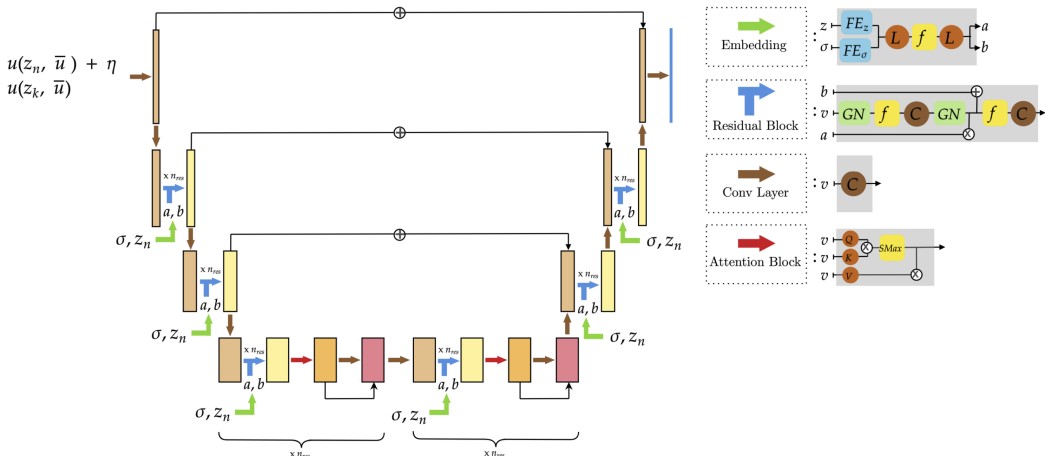

Figure 5: Architecture of the conditional UViT backbone used in our diffusion model.

upsampling paths are connected by skip connections, as in a standard U-Net, to preserve fine-scale information.

Conditioning on the diffusion time $\sigma$ and the latent code $z_n$ (green arrows in Fig. 5) is implemented via feature-wise affine modulation. The pair $(z_n, \sigma)$ is first passed through Fourier encoders $\mathrm{FE}_z$ and $\mathrm{FE}_\sigma$, followed by linear layers that produce scale and shift vectors $(a, b)$. Within each residual block, these vectors modulate the normalized features through a FiLM-style transformation $a \odot v + b$, where $v$ denotes the current feature map. Residual blocks consist of GroupNorm (GN), pointwise nonlinearities $f$, and convolutional layers $C$, as indicated on the right of Fig. 5. Attention blocks use standard multi-head self-attention with query–key–value projections $(Q, K, V)$ and a softmax operator, applied to the flattened spatial tokens at the corresponding resolution. This design combines the locality and multiscale structure of a U-Net with the long-range dependencies and flexible conditioning of transformers, providing a strong backbone for our diffusion model in 3D Helmholtz problems.

## C.2    3D RESULTS

We further validate our approach on fully 3D Helmholtz problems defined on $64^3$ grids, using four frequencies $5 \times 10^5$, $10^6$, $1.5 \times 10^6$, and $2.5 \times 10^6$ Hz. In this setting, we compare three models: a diffusion model with U-Net backbone (*Diffusion–UNet*), a diffusion model with UViT backbone (*Diffusion–UViT*), and a vanilla deterministic U-Net trained to predict the wavefield directly. The 3D results reproduce the main trends observed in 2D: relative $L^2$, $H^1$, and energy errors increase with frequency for all models, but Diffusion–UViT consistently achieves the lowest errors at all four frequencies, substantially outperforming both Diffusion–UNet and the vanilla U-Net. Qualitatively, the 3D volume renderings in Fig. 6 mirror these trends: the U-Net prediction tends to remain concentrated near the center of the domain, consistent with the 2D case, whereas both diffusion models better recover the wavefield near the boundaries, with Diffusion–UViT providing the sharpest boundary reconstruction.

Table 2: **3D Helmholtz errors on** $64^3$ **grids.** Relative $L^2$, $H^1$, and energy errors (mean $\pm$ std, clamped to $\leq 1$) over $N = 500$ test samples for our frequencies. Diffusion–UViT consistently outperforms Diffusion–UNet and the vanilla U-Net.

| Frequency (Hz) | Metric | Diffusion–UNet | Diffusion–UViT | U-Net only |
|---|---|---|---|---|
| $5 \times 10^5$ | $L^2$ | 0.030 | **0.013** | 0.180 |
| | $H^1$ | 0.050 | **0.019** | 0.287 |
| | Energy | 0.041 | **0.019** | 0.102 |
| $1.0 \times 10^6$ | $L^2$ | 0.032 | **0.011** | 0.295 |
| | $H^1$ | 0.049 | **0.016** | 0.455 |
| | Energy | 0.037 | **0.012** | 0.170 |
| $1.5 \times 10^6$ | $L^2$ | 0.063 | **0.016** | 0.874 |
| | $H^1$ | 0.091 | **0.024** | 1.000 |
| | Energy | 0.101 | **0.014** | 0.752 |
| $2.5 \times 10^6$ | $L^2$ | 0.159 | **0.078** | 0.967 |
| | $H^1$ | 0.229 | **0.112** | 1.000 |
| | Energy | 0.178 | **0.049** | 0.793 |

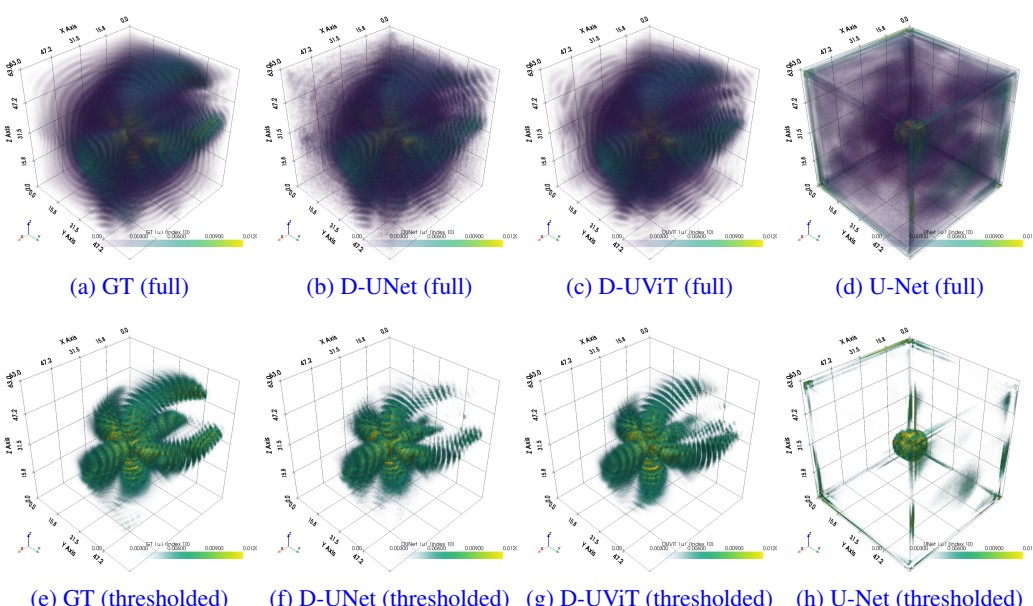

(a) GT (full)   (b) D-UNet (full)   (c) D-UViT (full)   (d) U-Net (full)

(e) GT (thresholded)   (f) D-UNet (thresholded)   (g) D-UViT (thresholded)   (h) U-Net (thresholded)

Figure 6: Example 3D Helmholtz wavefield visualization using volume rendering from a fixed viewpoint: top row shows the full 3D visualization; bottom row uses a threshold to focus on center region to elaborate the differences.

Table 3: **Model size, wall-clock evaluation time, and accuracy on 3D Helmholtz data.** Parameter counts are total learnable parameters; times are wall-clock for one solve; errors are relative.

| Method | # Params | Wall-clock time (s) | $\mathcal{E}_{L^2}$ | $\mathcal{E}_{H^1}$ | $\mathcal{E}_{\text{energy}}$ |
|---|---|---|---|---|---|
| U-Net | 1,883,969 | 0.006 | 0.967 | 1.000 | 0.793 |
| Diffusion–U-Net | 1,947,153 | 3.142 | 0.159 | 0.229 | 0.178 |
| Diffusion–UViT | 6,781,555 | 5.150 | **0.078** | **0.112** | **0.049** |
| Direct solver (J-Wave) | — | 68.040 | — | — | — |

# D ADDITIONAL RESULTS

This section provides extended qualitative and quantitative results that complement the main text.

## D.1 DATA EXAMPLE

Figure 7 shows representative input–output pairs across frequencies: as the driving rate increases, the Helmholtz solution exhibits denser interference fringes and faster phase oscillations. Using angular frequency $\omega = 2\pi f$ and local sound speed $c(x)$, the wavelength is

$$\lambda(x) \;=\; \frac{2\pi\,c(x)}{\omega}.$$

With an average speed $\bar{c} \approx 2000$ (simulation units) and our largest angular frequency $\omega_{\max} = 2.5 \times 10^6$ rad/s, the shortest wavelength is

$$\lambda_{\min} \;=\; \frac{2\pi\,\bar{c}}{\omega_{\max}} \;=\; \frac{2\pi \times 2000}{2.5 \times 10^6} \;\approx\; 5 \times 10^{-3}.$$

We use a uniform grid with $\Delta x = \Delta y = 10^{-3}$, yielding $\lambda_{\min}/\Delta x \approx 5$ samples per shortest wavelength, which is adequate to resolve the fine oscillatory structure in the ground-truth wavefields.

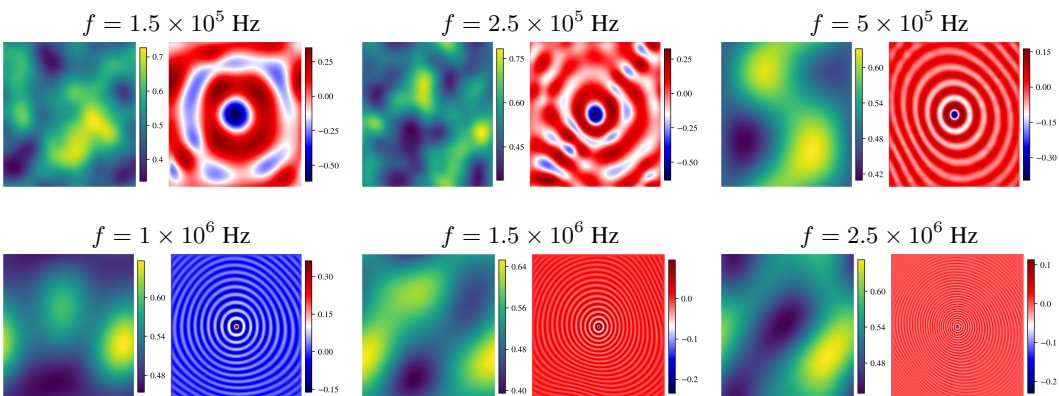

Figure 7: **Input–output data pairs by frequency.** Each tile shows one random test example: left—coefficient field $c$; right—solution $P$. Color scales are set per panel.

## D.2 SAMPLER ABLATION

For completeness we describe the three samplers used in our ablation, in the same notation as the main text. Let $\{\beta_t\}_{t=1}^{T}$ be a discrete noise schedule (linear or cosine), $\alpha_t := 1 - \beta_t$, and $\bar{\alpha}_t := \prod_{s=1}^{t} \alpha_s$. The forward process is $q(z_t \mid z_0) = \mathcal{N}(\sqrt{\bar{\alpha}_t}\,z_0, (1 - \bar{\alpha}_t)I)$ and the network predicts noise $\varepsilon_\theta(z_t, t, c)$ given the condition $c$. DDPM (ancestral): We use the standard Gaussian reverse transition

$$p_\theta(z_{t-1} \mid z_t, c) = \mathcal{N}\Big(z_{t-1};\, \mu_\theta(z_t, t, c),\, \tilde{\beta}_t I\Big),$$

$$\mu_\theta(z_t, t, c) = \frac{1}{\sqrt{\alpha_t}} \left( z_t - \frac{\beta_t}{\sqrt{1 - \bar{\alpha}_t}}\, \varepsilon_\theta(z_t, t, c) \right). \tag{39}$$

with posterior variance $\tilde{\beta}_t = \frac{1 - \bar{\alpha}_{t-1}}{1 - \bar{\alpha}_t}\, \beta_t$. We ablate *linear* and *cosine* schedules and step budgets $T \in \{10, 50, 100, 1000\}$. Over $T$ steps, the update draws $z \sim \mathcal{N}(0, I)$ at each step ($t > 1$) and sets $z_{t-1} = \mu_\theta + \sigma_t z$ with $\sigma_t = \sqrt{\tilde{\beta}_t}$. DDIM (implicit): Using the same schedule, define the predicted clean sample $\hat{z}_0(z_t, t, c) = \frac{1}{\sqrt{\bar{\alpha}_t}}\big(z_t - \sqrt{1 - \bar{\alpha}_t}\,\varepsilon_\theta(z_t, t, c)\big)$. For a decreasing sequence $t_1 > t_2 > \cdots > t_S$, DDIM updates deterministically ($\eta = 0$) as

$$z_{t_{i+1}} \;=\; \sqrt{\bar{\alpha}_{t_{i+1}}}\,\hat{z}_0 \;+\; \sqrt{1 - \bar{\alpha}_{t_{i+1}}}\, \varepsilon_\theta(z_{t_i}, t_i, c), \tag{40}$$

Table 4: **Sampler–step ablation at** $f = 2.5 \times 10^6$ **Hz** (relative error; lower is better). Cosine schedule unless noted. Values rounded to 3 decimals; best per metric in **bold**.

| | $\mathcal{E}_{L^2} \downarrow$ | | | | $\mathcal{E}_{H^1} \downarrow$ | | | | $\mathcal{E}_{\text{energy}} \downarrow$ | | | |
|---|---|---|---|---|---|---|---|---|---|---|---|---|
| **Sampler** | **10** | **50** | **100** | **1000** | **10** | **50** | **100** | **1000** | **10** | **50** | **100** | **1000** |
| DDPM (linear) | 0.999 | 0.956 | 0.942 | 0.918 | 0.999 | 0.797 | 0.720 | 0.631 | 0.999 | 0.949 | 0.932 | 0.901 |
| DDPM (cosine) | 0.457 | 0.268 | 0.234 | **0.188** | 0.180 | 0.106 | 0.095 | **0.078** | 0.138 | 0.100 | 0.093 | **0.082** |
| DDIM (cosine) | 0.470 | 0.333 | 0.305 | 0.284 | 0.189 | 0.139 | 0.129 | 0.122 | 0.137 | 0.122 | 0.124 | 0.125 |
| SDE (cosine) | 0.458 | 0.373 | 0.358 | 0.336 | 0.190 | 0.151 | 0.146 | 0.138 | 0.168 | 0.126 | 0.124 | 0.129 |

Table 5: **Sampler–step ablation at** $f = 2.5 \times 10^6$ **Hz** (wall-clock time per 500 samples in seconds).

| | Time (s) | | | |
|---|---|---|---|---|
| **Sampler** | **10** | **50** | **100** | **1000** |
| DDPM (linear) | 8.927 | 46.892 | 94.534 | 948.698 |
| DDPM (cosine) | 19.541 | 57.521 | 95.430 | 951.771 |
| DDIM (cosine) | 10.041 | 47.731 | 95.023 | 948.228 |
| SDE (cosine) | 9.752 | 47.713 | 94.918 | 947.771 |

or stochastically with $\eta \in [0, 1]$ by decomposing the second term into an $\eta z$ component plus a $(1 - \eta^2)^{1/2} \varepsilon_\theta$ component. In our ablation we use the *cosine* schedule and report the $\eta=0$ (deterministic) case. Score-based SDE (continuous-time): In the variance-preserving formulation $dz = -\frac{1}{2}\beta(t)\, z\, dt + \sqrt{\beta(t)}\, dW_t$, the closed-form mean/variance maps from $t$ to $t - \Delta t$ can be written as $z_{t-\Delta t} = r(t, \Delta t)\, z_t + \big(\sigma(t-\Delta t) - r(t, \Delta t)\sigma(t)\big)\, \varepsilon_\theta(z_t, t, c)$, where $r(t, \Delta t) = \mu(t - \Delta t)/\mu(t)$ and $(\mu, \sigma)$ are the analytic solution scalings of the VP–SDE[1]. This implements an ancestral sampler in continuous time, and we use the *cosine* schedule for $\beta(t)$ via its discretized counterpart and the same step budgets as above. Across methods, DDPM introduces stepwise Gaussian noise (higher diversity, slightly higher variance at small $T$), DDIM provides a deterministic path given the same $\varepsilon_\theta$ (lower variance and faster convergence at small $T$), and the SDE sampler follows the continuous-time ancestral update governed by $(\mu, \sigma)$. Our ablation (Appendix Table 4) compares these choices under identical networks and conditions.

In addition to accuracy, we also report the wall-clock time required to generate a single sample for each sampler and step budget in Tab. 5. As discussed in the main text, our current sampling procedure is computationally expensive. Recent work has proposed faster ODE-based samplers and exponential-integrator (EI) schemes for diffusion models—for example, the exponential integrator method of Zhang & Chen (2022)—which we plan to explore in future work. For all remaining experiments, we therefore adopt a cosine-schedule DDPM sampler with 1000 steps as our default, as it offers the best accuracy in the high-frequency Helmholtz regime.

To contextualize this choice, Tab. 6 compares parameter counts and wall-clock time per sample across the baseline models and our chosen diffusion configuration. Even with 1000 denoising steps, the diffusion model remains within a practical runtime range relative to the deterministic neural operators, and still operates orders of magnitude faster than the numerical Helmholtz solver, which requires several minutes of computation per sample on the same hardware.

Table 6: **Model size, evaluation cost, and accuracy at** $f = 2.5 \times 10^6$ **Hz.** Parameter counts are total learnable parameters; times are wall-clock evaluation times for a single sample; errors are relative. The direct J-Wave solver is treated as the reference with zero relative error.

| Model | # Parameters | Eval. time (s) | $\mathcal{E}_{L^2}$ | $\mathcal{E}_{H^1}$ | $\mathcal{E}_{\text{energy}}$ |
|---|---|---|---|---|---|
| U-Net | 91.9M | 0.01 | 0.767 | 1.000 | 0.514 |
| FNO | 15.2M | 0.07 | 0.412 | 0.494 | 0.141 |
| HNO | 73.1M | 0.15 | 0.802 | 0.996 | 0.590 |
| Diffusion (DDPM/1000) | 92.6M | 1.83 | **0.095** | **0.135** | **0.036** |
| Direct solver (J-Wave) | — | 23.84 | — | — | — |

---

[1]In code, `mu()` and `sigma()` implement these scalings, and we integrate $t$ linearly from $1 \rightarrow 0$ with step $1/T$.

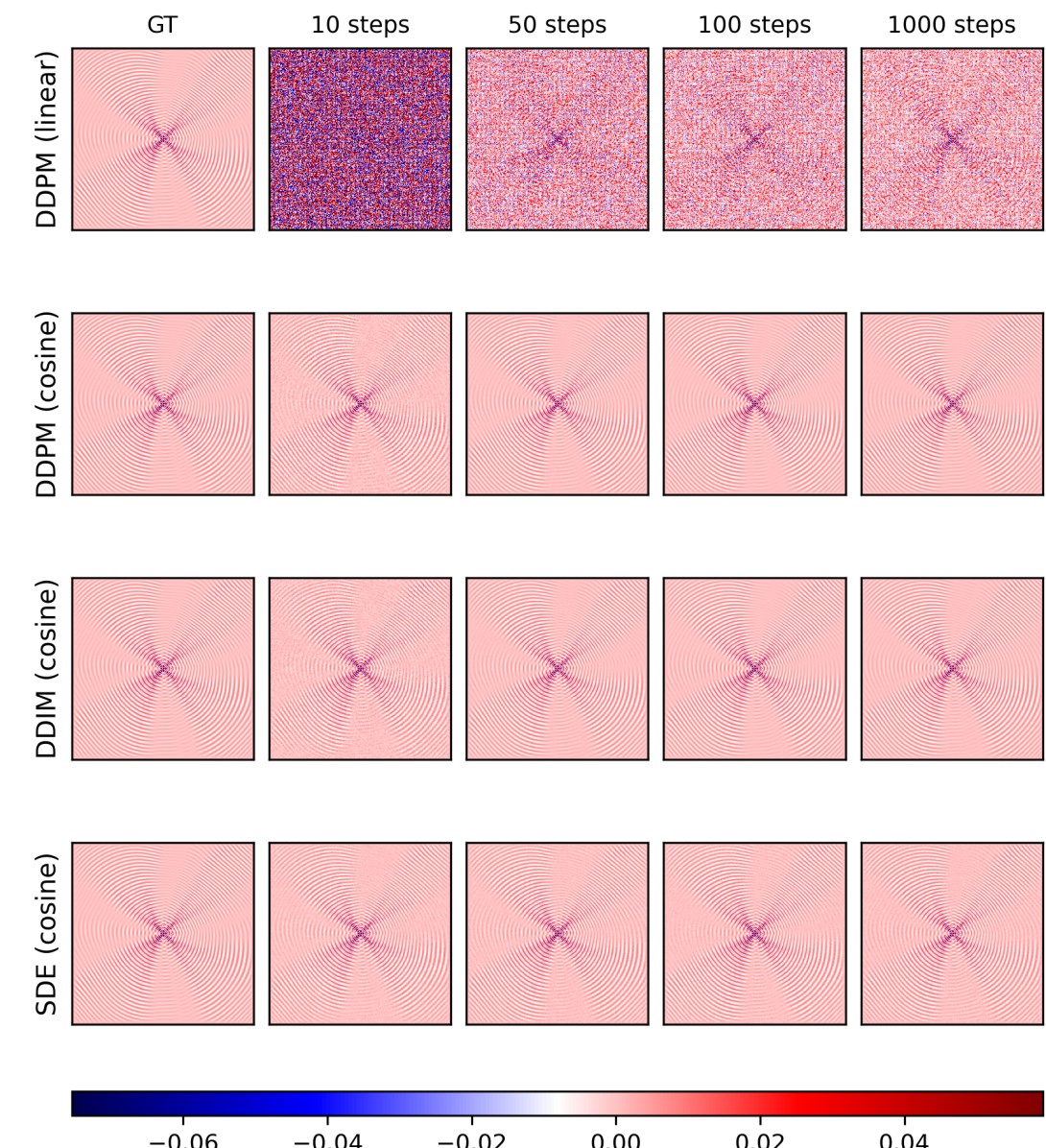

Figure 8: Comparison of sampler behavior across step budgets at $f = 2.5 \times 10^6$ Hz.

### D.3 ADDITIONAL QUALITATIVE RESULTS AND SPECTRAL ANALYSIS

Here we provide an expanded set of visual comparisons across all evaluated frequencies and multiple randomly drawn sound–speed maps (see App. Figs. 14–19). For each case, we show GT, U-Net, FNO, HNO, and Diffusion predictions alongside per-pixel error maps. The trends observed in the main text persist: the Diffusion model consistently resolves interference patterns and high-frequency details with reduced artifacts, while deterministic operators exhibit smoothing and phase misalignment—especially in far-field regions and at higher $k$.

To make these observations more quantitative, we also analyze the 2D Fourier power spectra of the predicted wavefields. For a fixed high frequency (e.g., $f = 2.5 \times 10^6$ Hz), we compute the 2D FFT of GT and each model prediction and visualize the corresponding power $|F(k_x, k_y)|^2$ over the full $(k_x, k_y)$ domain. As illustrated in Fig. 9, the deterministic baselines (in particular the U-Net backbone and HNO) exhibit clear spectral bias: they oversmooth or misallocate energy at large wavenumbers, deviating substantially from the ground-truth spectrum. In contrast, the Diffusion model closely

tracks the GT power spectrum across both low and high spatial frequencies, indicating that it captures the full range of oscillatory content rather than collapsing toward low-$k$ modes.

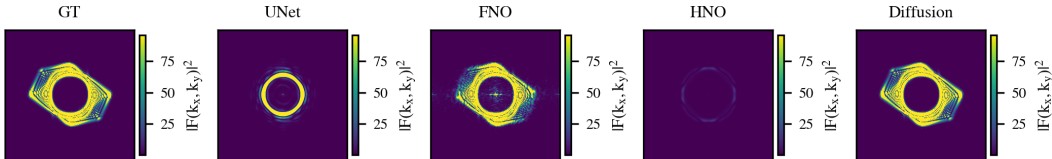

Figure 9: **2D spectral power comparison at** $f = 2.5 \times 10^6$ **Hz.** Example Fourier power spectra $|F(k_x, k_y)|^2$ for GT and each model on a single test sample.

Table 7: **1D relative errors vs. frequency.** Diffusion reports mean±std over $K{=}10$ samples.

| Freq (Hz) | $L^2$ | | $H^1$ | | Energy | |
|---|---|---|---|---|---|---|
| | **Diffusion** | **U-Net** | **Diffusion** | **U-Net** | **Diffusion** | **U-Net** |
| 1.5e5 | **0.022**±0.0003 | 0.065 | **0.036**±0.0003 | 0.108 | **0.014**±0.0001 | 0.032 |
| 2.5e5 | **0.030**±0.001 | 0.087 | **0.046**±0.001 | 0.130 | **0.010**±0.0002 | 0.025 |
| 5e5 | **0.091**±0.002 | 0.174 | **0.114**±0.002 | 0.242 | **0.011**±0.0003 | 0.044 |
| 7.5e5 | **0.093**±0.004 | 0.295 | **0.117**±0.004 | 0.389 | **0.018**±0.001 | 0.096 |
| 1e6 | **0.215**±0.003 | 0.395 | **0.264**±0.004 | 0.510 | **0.050**±0.002 | 0.181 |

### D.4 COMPLETE SENSITIVITY PANELS (NEAR/FAR)

To quantify input–output sensitivity, we construct a perturbation-driven dataset around a fixed reference field $c_0$. We sample $D = 100$ independent Gaussian random–field (GRF) realizations $\{c^{(d)}\}_{d=1}^D$ and, for each direction $d$, define a *fixed direction* in coefficient space as the straight line

$$c^{(d)}(s) = (1-s)\, c_0 + s\, c^{(d)}, \qquad s \in [0,1],$$

which linearly connects the common reference $c_0$ ($s{=}0$) to a particular perturbation $c^{(d)}$ ($s{=}1$). We solve the Helmholtz problem at 100 uniformly spaced $s$-values along every line, yielding trajectories that move from $c_0$ toward increasingly perturbed media. A schematic in *coefficient space* is provided in Figure 10: black markers indicate the sampled endpoints $\{c^{(d)}\}$, while gray contours illustrate intermediate states $c^{(d)}(s)$ for $s \in \{0.3, 0.5, 0.7, 1.0\}$, contracting smoothly toward $c_0$.

**Sampling along a fixed direction.** For a chosen spatial probe (pixel) $x^*$, a *sampling curve* plots the wavefield amplitude $u(x^*; c^{(d)}(s))$ as a function of $s \in [0,1]$ for a single direction $d$. This reveals how the output at $x^*$ responds as the input medium moves *along one line* in coefficient space. In App. Figs. 20–**??** we show these curves for eight probes (4 *near* the source and 4 *far* near the PML boundary) and for different directions.

**Kernel density estimate (KDE) across directions.** To summarize variability *across directions* at a fixed interpolation level $s$, we form the set $\{u(x^*; c^{(d)}(s))\}_{d=1}^D$ for a probe $x^*$ and estimate its distribution with a one-dimensional kernel density estimator. Writing the scalar quantity of interest as $y_d = g(u(x^*; c^{(d)}(s)))$, the KDE is

$$\widehat{p}_s(y) = \frac{1}{Dh} \sum_{d=1}^D K\left(\frac{y - y_d}{h}\right),$$

with kernel $K$ (Gaussian in our plots) and bandwidth $h{>}0$. The KDE panel therefore shows how dispersed (broad) or concentrated (narrow) the responses are across GRF directions at a given $s$. Broad, multi-modal densities indicate amplified sensitivity and interference variability, while narrow peaks indicate low variability. We report KDEs at representative $s$-values (e.g., $s{=}0$, $0.1$, and $1$) for both *near* and *far* probes to contrast regimes of low vs. high sensitivity (see App. Figs. 22–24).

To verify that these observations are not an artifact of a single frequency, we also perform the sensitivity analysis at lower frequencies. App. Fig. 25 shows the KDEs at $s{=}0$ for $1.5 \times 10^5$, $5 \times 10^5$, $10^6$, and $1.5 \times 10^6$ Hz. Across all four frequencies we observe the same qualitative trend as at

the highest frequency: deterministic operators produce narrow, highly concentrated densities that under-represent variability across GRF directions, whereas the diffusion model maintains broader, multi-modal responses, indicating preserved sensitivity to small perturbations in the coefficient field even in easier (lower-frequency) regimes.

In summary, the *sampling* plots diagnose *direction-wise* response along a line in coefficient space, while the *density* (KDE) plots aggregate *across directions* at fixed $s$ to characterize variability and calibration of the predictive models.

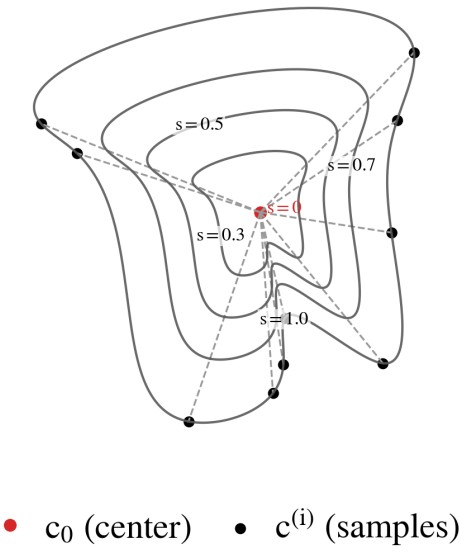

• $c_0$ (center)   • $c^{(i)}$ (samples)

Figure 10: **Illustration of unit ball in function space of coefficient fields.**

The 1D experiment reproduces the main 2D trends. We report relative $L^2$, $H^1$, and energy errors across frequencies in Table 7, comparing only the diffusion model and a backbone-matched U-Net. As frequency increases, errors rise for both models, but the diffusion model consistently maintains a clear advantage, including at the highest tested frequency ($10^6$ Hz). Sensitivity diagnostics mimic the 2D setup: at $f=10^6$ Hz the *cross-sampling* trajectories and *direction-wise density* plots (Figures 11 and 12) show that diffusion tracks the ground-truth path along $s$ and recovers the broadened amplitude distribution across directions, while the U-Net under-responds and tends toward collapse in the far field.

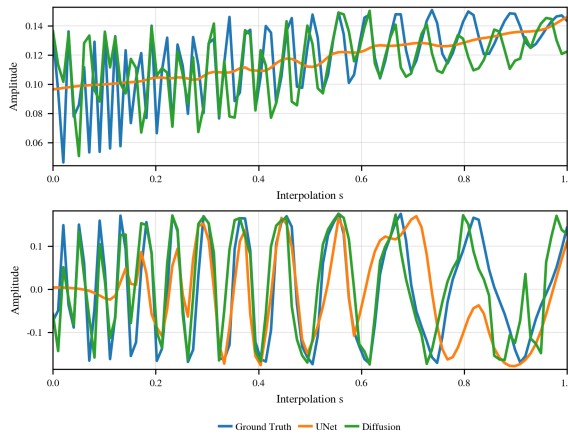

Figure 11: **Sampling along a linear path in coefficient space (1D).** Example curves from $c_0$ to the target sound-speed map along direction $d=10$ with $s \in [0, 1]$ (100 samples). *Top:* near the source. *Bottom:* far from the source.

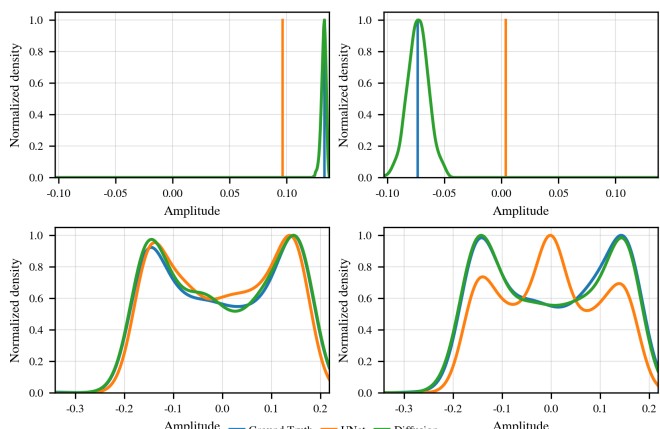

Figure 12: **Density plot (1D).** Kernel density estimates of $\{ u_M(s,d;y,x) \}_{d=1}^{100}$ at $s = 0$ (top) and $s = 1$ (bottom), shown for a *near* point (left) and a *far* point (right). At $s = 0$ all inputs equal $c_0$, so deterministic models collapse to a spike, whereas diffusion exhibits calibrated spread; at $s \approx 1$, all methods broaden, with diffusion better capturing the high-variance regime.

Beyond pointwise trajectories, we summarize sensitivity by an *average directional variance* curve: for each $s$, we compute the variance across directions $d$ of the predicted amplitude at every pixel and then average over the domain. As frequency increases, the domain-averaged variance shrinks, making the target curve increasingly subtle; deterministic surrogates (U-Net, FNO, HNO) fail to resolve these small variations and tend to under-curve, whereas the diffusion operator continues to track the ground-truth trajectory across $s$, including the late sharp rise toward $s{=}1$ (Figure 13).

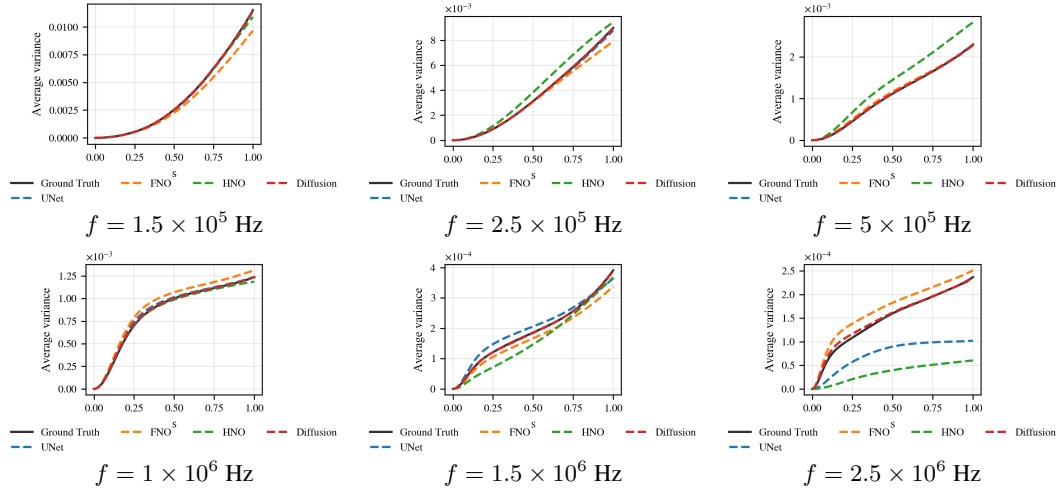

Figure 13: **Average directional variance vs. interpolation $s$ across frequencies.** Each panel shows domain-averaged variance across 100 directions as a function of $s$.

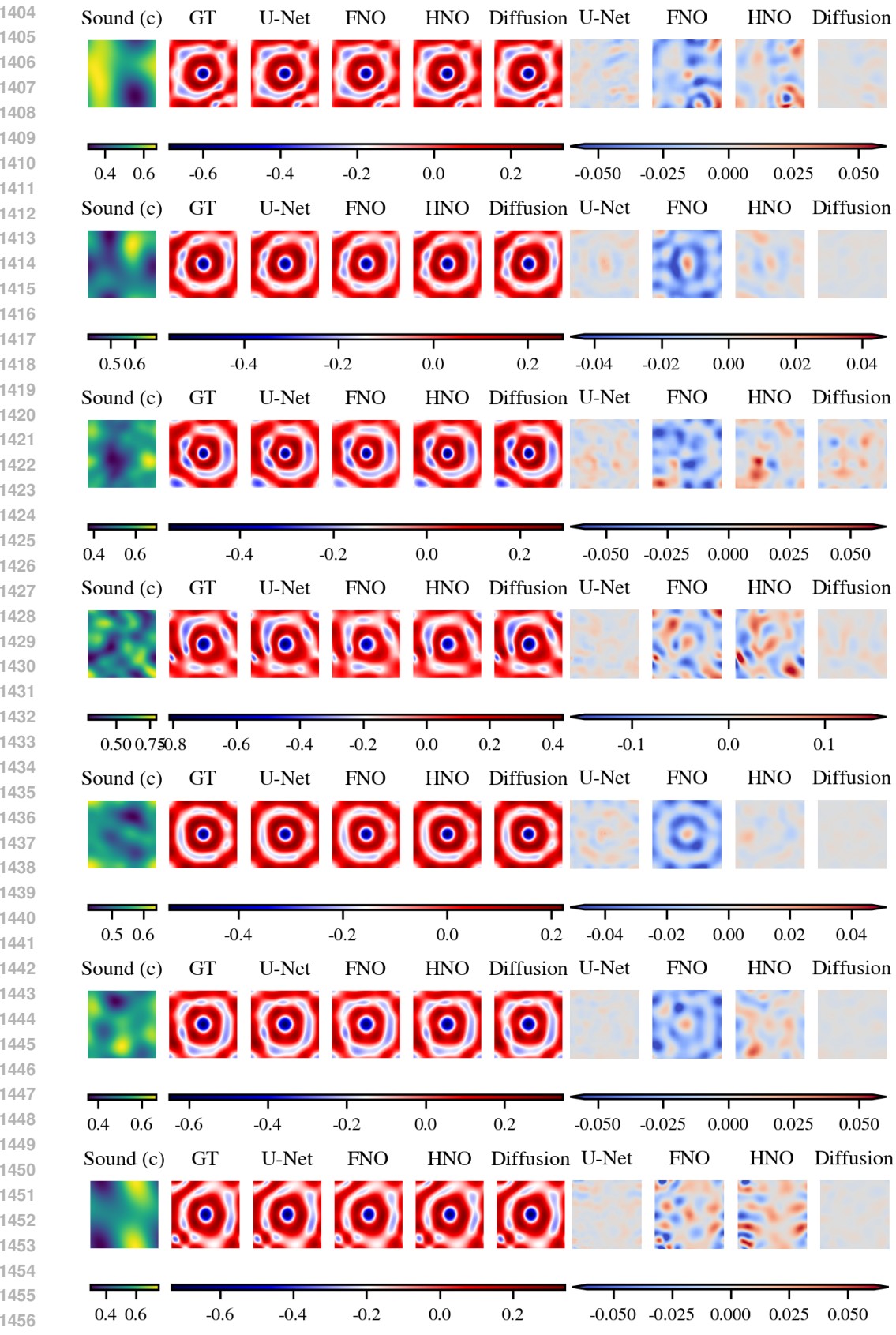

*Prediction vs. Ground Truth*        *Residual (Pred−GT)*

Figure 14: **Qualitative comparisons at** $f = 1.5 \times 10^5$ **Hz.**

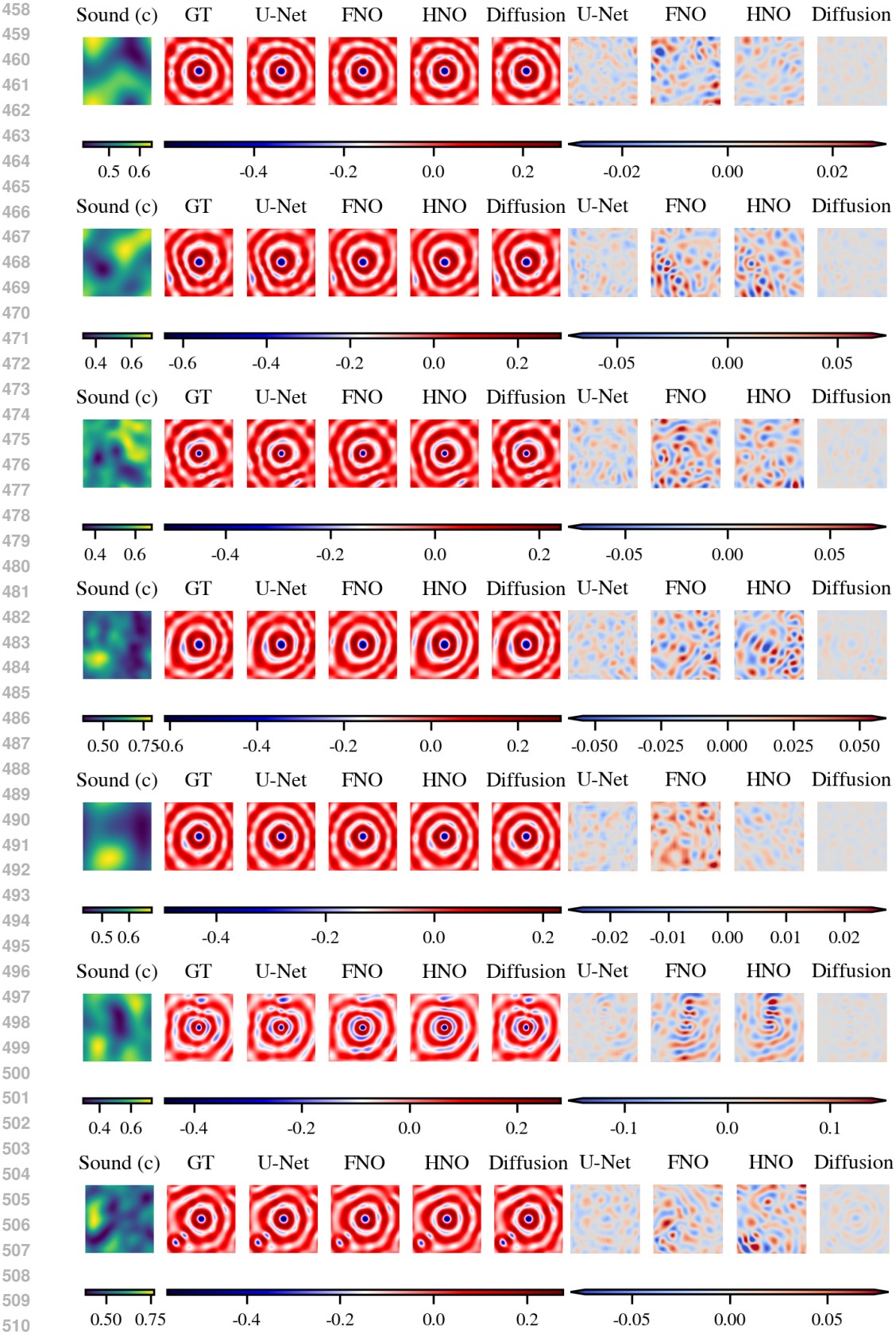

*Prediction vs. Ground Truth*          *Residual (Pred−GT)*

Figure 15: **Qualitative comparisons at** $f = 2.5 \times 10^5$ **Hz.**

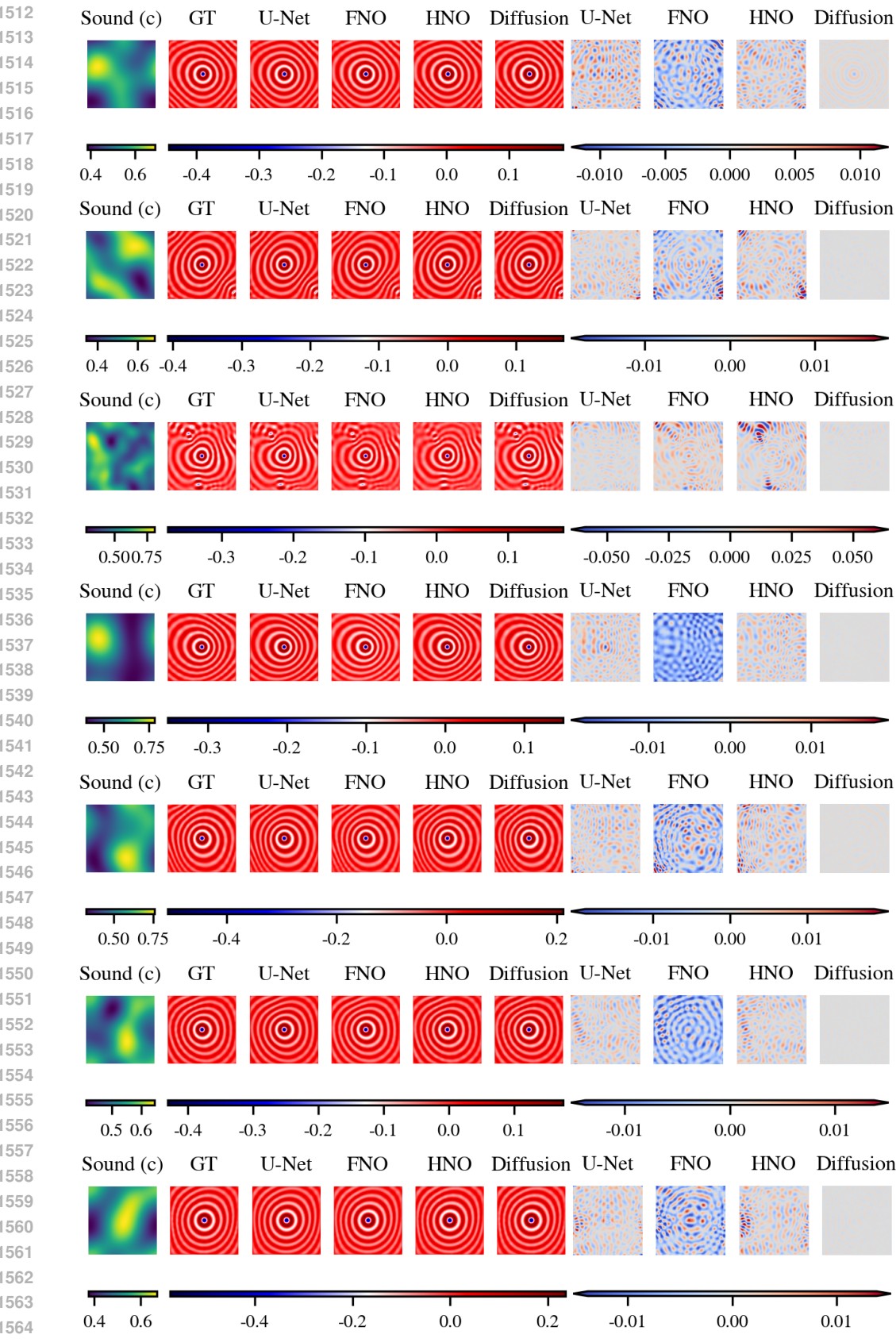

*Prediction vs. Ground Truth*              *Residual (Pred−GT)*

Figure 16: **Qualitative comparisons at** $f = 5 \times 10^5$ **Hz.**

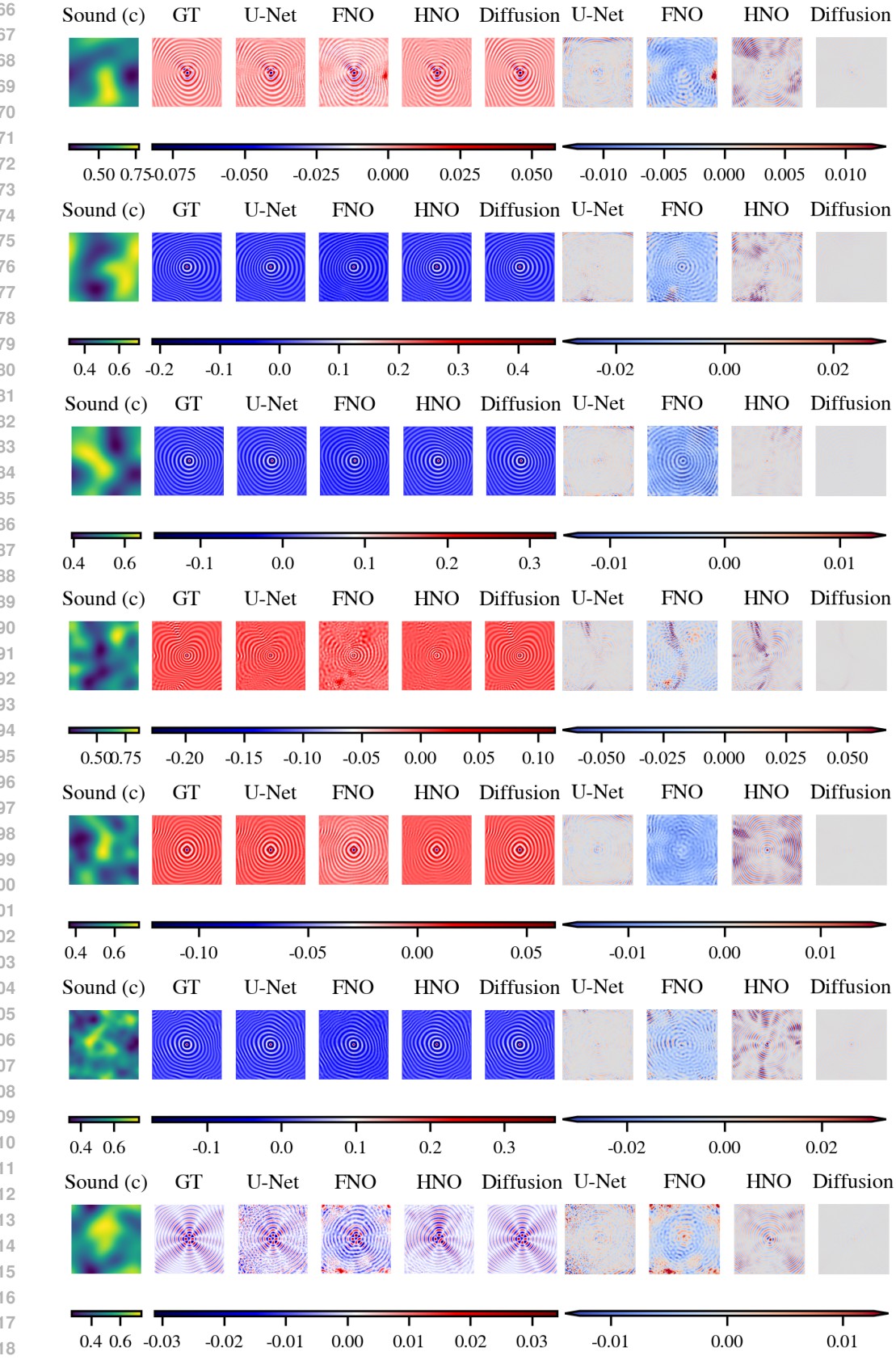

*Prediction vs. Ground Truth*  *Residual (Pred−GT)*

Figure 17: **Qualitative comparisons at** $f = 1 \times 10^6$ **Hz.**

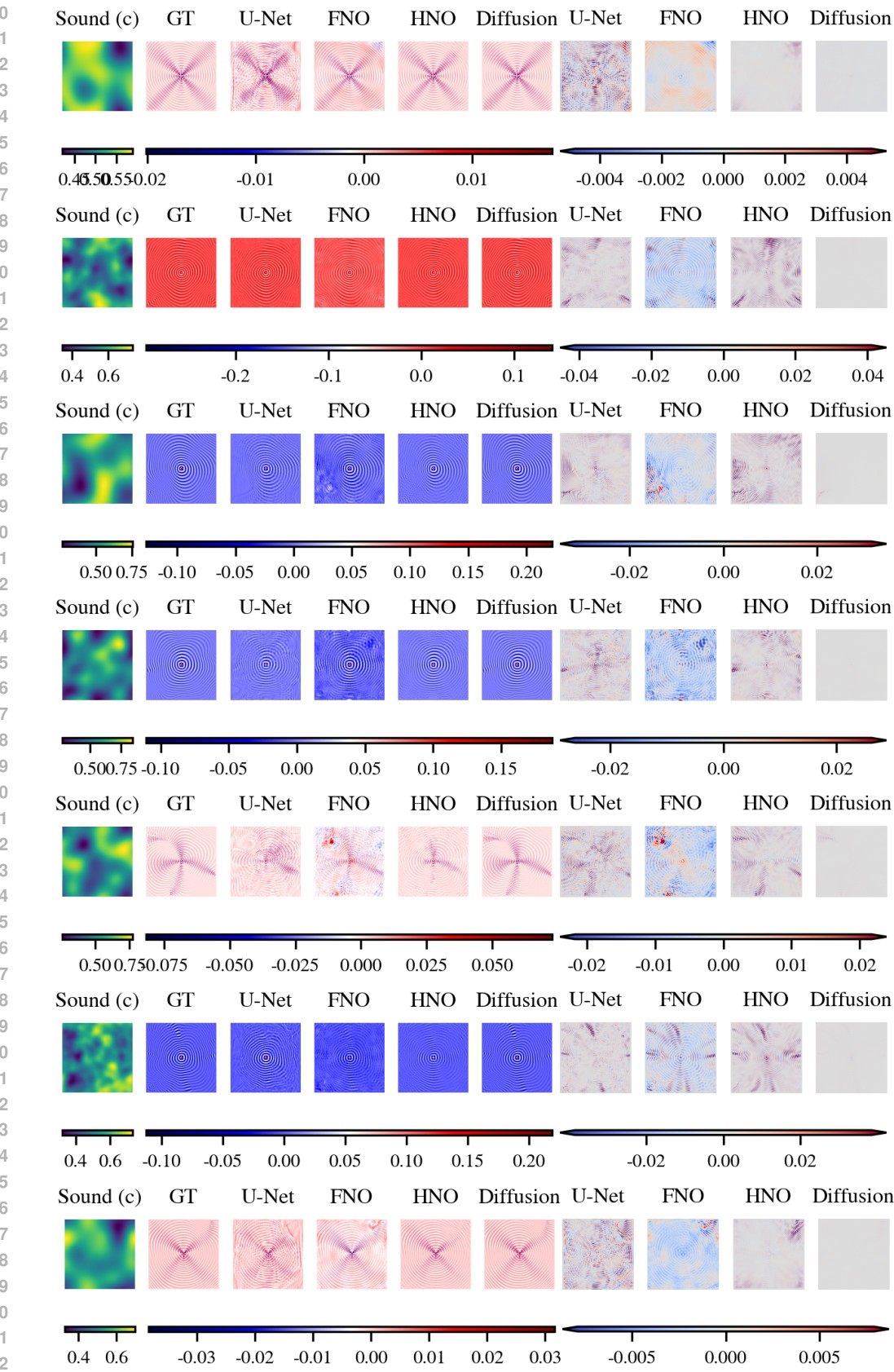

*Prediction vs. Ground Truth*      *Residual (Pred−GT)*

Figure 18: **Qualitative comparisons at** $f = 1.5 \times 10^6$ **Hz.**

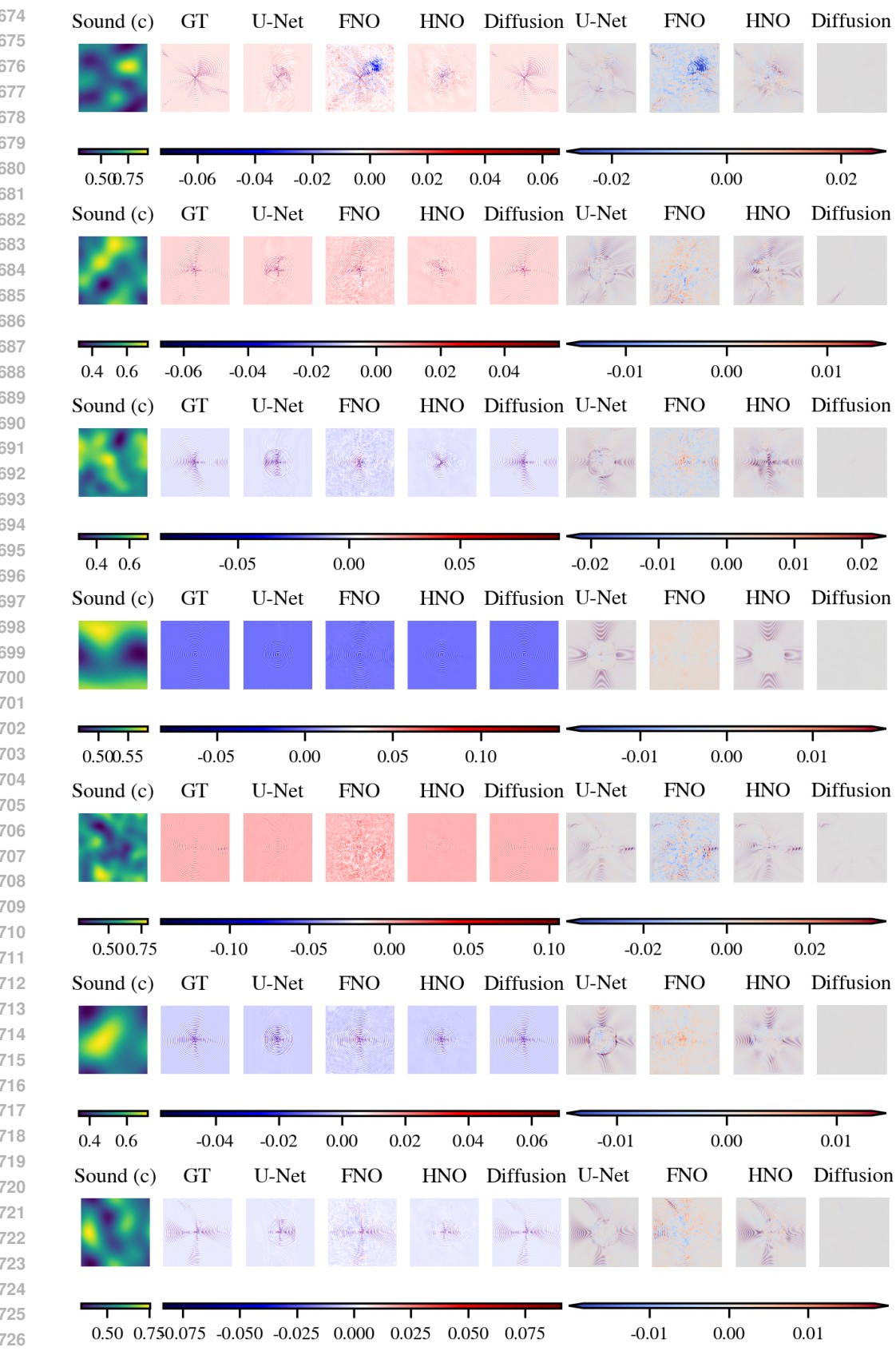

*Prediction vs. Ground Truth*          *Residual (Pred−GT)*

Figure 19: **Qualitative comparisons at** $f = 2.5 \times 10^6$ **Hz.**

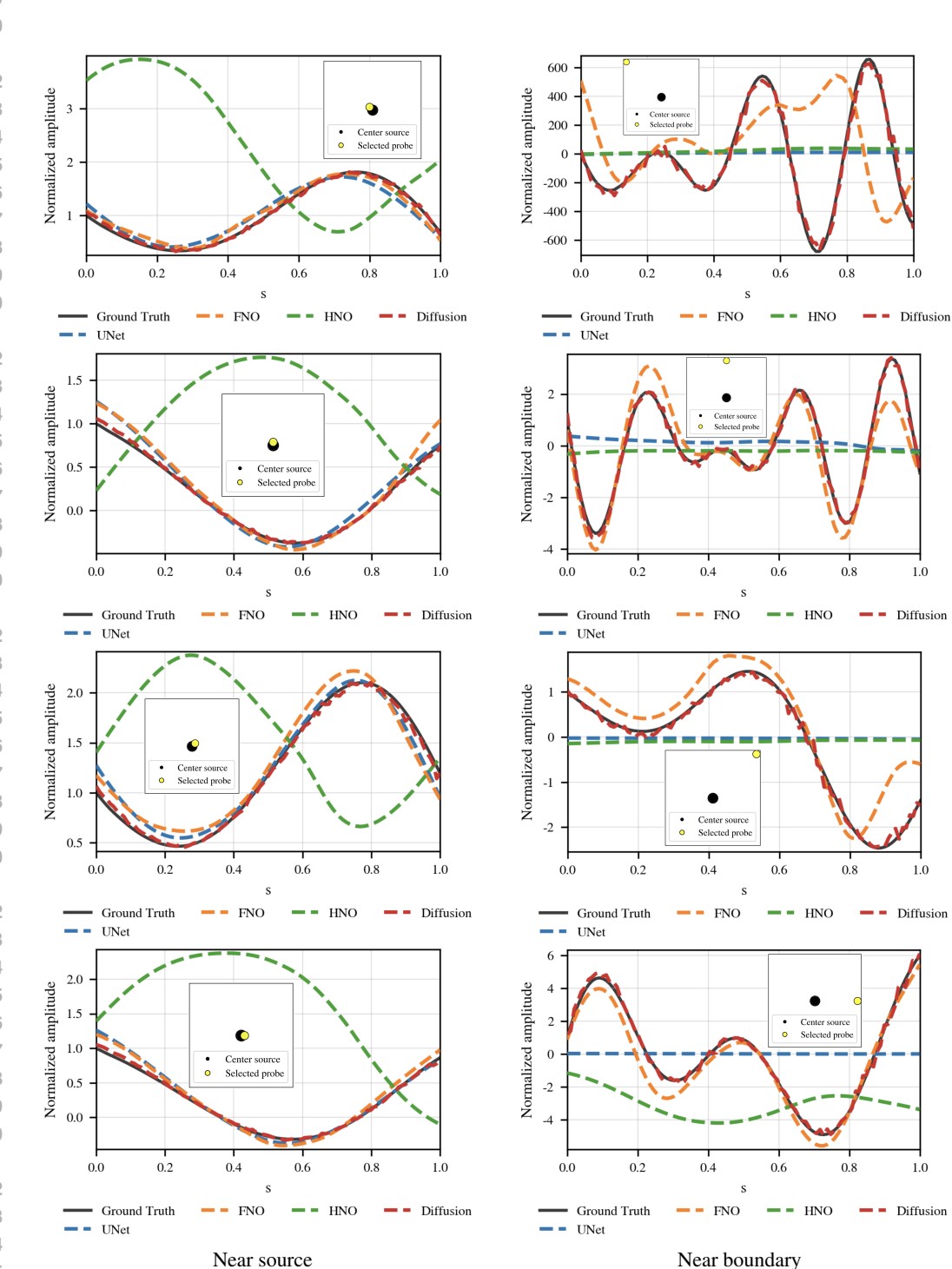

Figure 20: **Sampling along a coefficient path d=1 (all 4 near vs. all 4 far).** For each pair of media, we linearly interpolate the sound speed $c_s = (1 - s)c_0 + s\,c_1$ with $s \in [0, 1]$ and track the wavefield amplitude at fixed probe pixels. Left column: near-source probes. Right column: near-boundary probes.

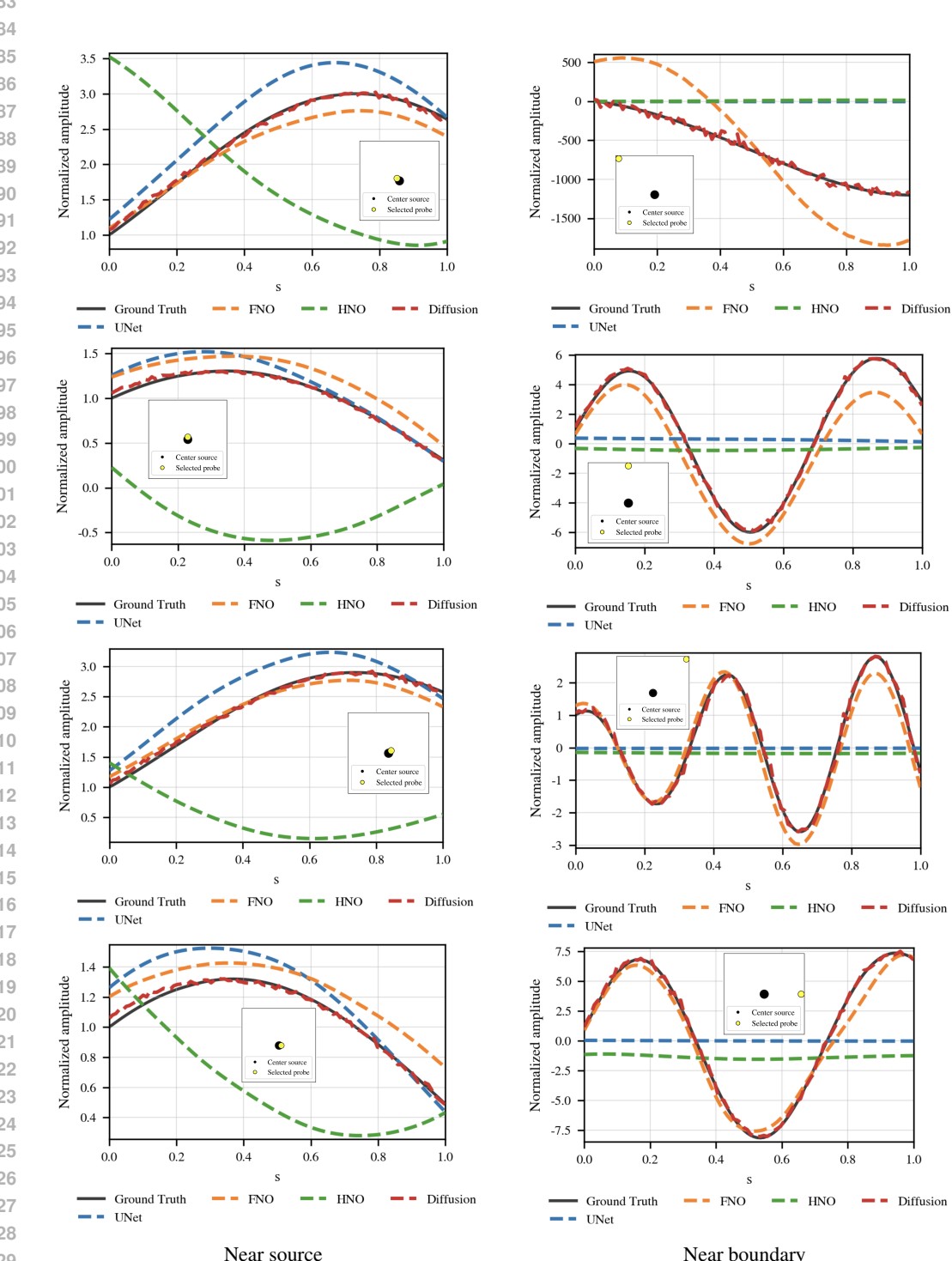

Near source                                    Near boundary

Figure 21: **Sampling along a coefficient path** d$=$**50 (all 4 near vs. all 4 far).** For each pair of media, we linearly interpolate the sound speed $c_s = (1-s)c_0 + s\,c_1$ with $s \in [0,1]$ and track the wavefield amplitude at fixed probe pixels. Left column: near-source probes. Right column: near-boundary probes.

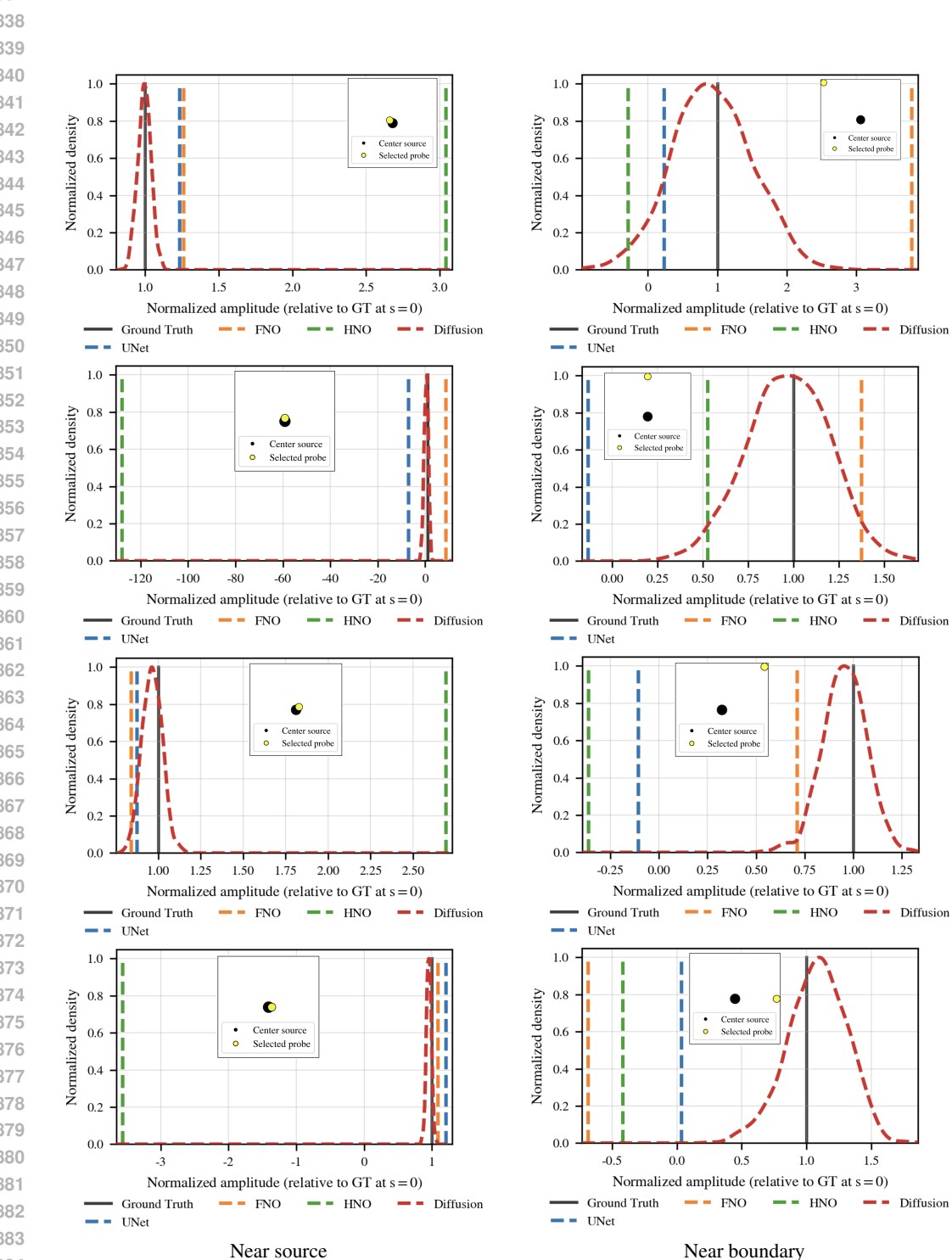

Near source                                                Near boundary

Figure 22: **Kernel density estimates across directions for s=0 (all 4 near vs. all 4 far).** Left column: near the source. Right column: near the boundary.

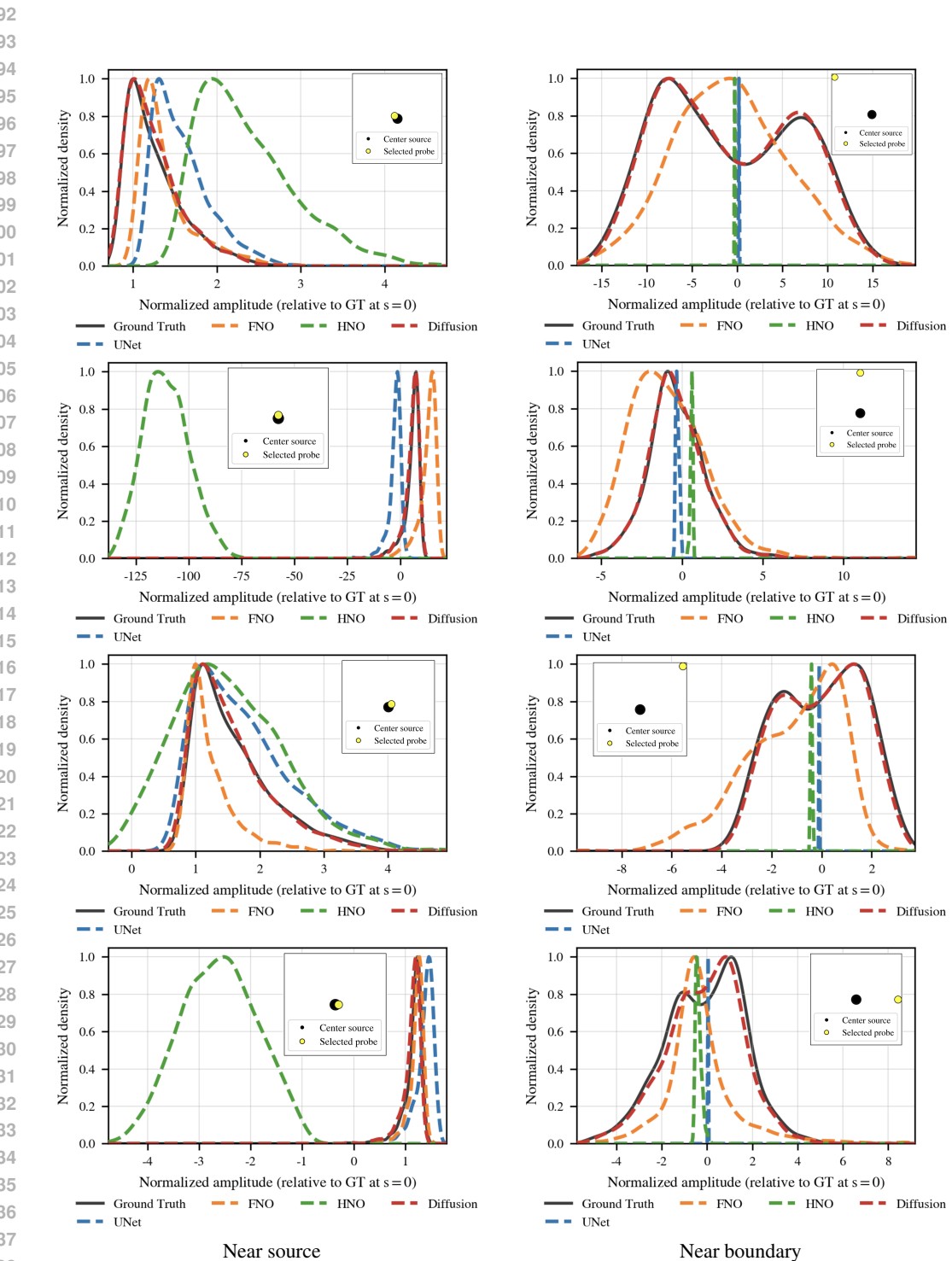

Near source                                              Near boundary

Figure 23: **Kernel density estimates across directions for s=0.1 (all 4 near vs. all 4 far).** Left column: near the source. Right column: near the boundary.

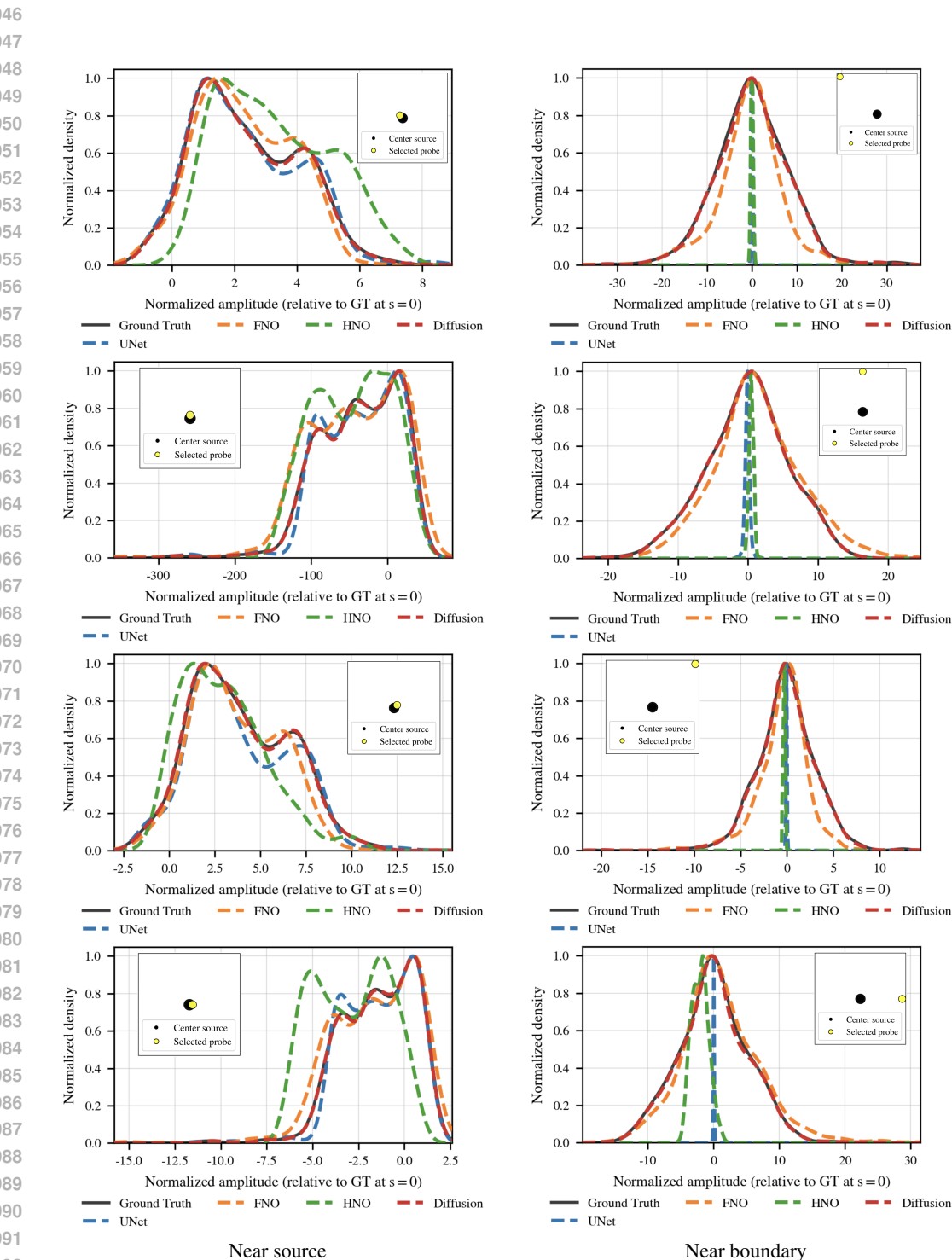

Near source                        Near boundary

Figure 24: **Kernel density estimates across directions for** $s=1$ **(all 4 near vs. all 4 far).** Left column: near the source. Right column: near the boundary.

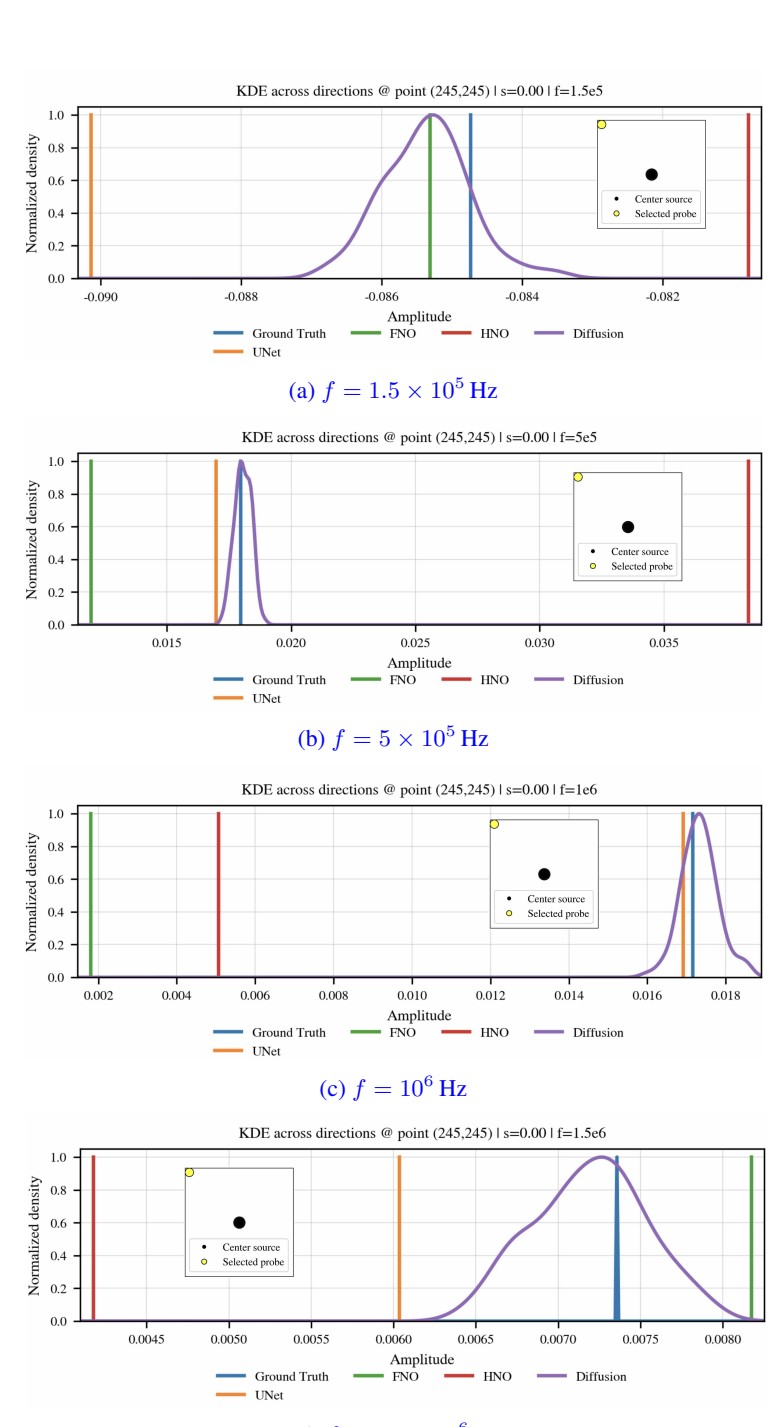

(a) $f = 1.5 \times 10^5$ Hz

(b) $f = 5 \times 10^5$ Hz

(c) $f = 10^6$ Hz

(d) $f = 1.5 \times 10^6$ Hz

Figure 25: KDEs of sensitivity responses at $s=0$ across GRF directions for four frequencies. At all frequencies, deterministic operators yield narrow, concentrated densities, while the diffusion model exhibits broader, multi-modal responses, indicating preserved sensitivity to small perturbations in the coefficient field.

## LLM USAGE DISCLOSURE

In accordance with ICLR policy, we disclose our limited use of large language models (LLMs) during manuscript preparation. LLMs (e.g. ChatGPT) were used *only* as general-purpose writing assistants for copy-editing and presentation polish, including: tightening grammar and style, clarifying phrasing, and harmonizing notation descriptions. LLMs were *not* used for research ideation, problem formulation, method design, data generation, implementation, experiments, analysis, or result interpretation.

