# OpenReview forum: "A Probabilistic Framework For Solving High-Frequency Helmholtz Equation Via Diffusion Models"
_ICLR.cc/2026/Conference — Submitted to ICLR 2026_

### Official Review · Reviewer_HF5E · 2025-10-24

[review text omitted: it was posted to a different submission]

---

> ### Author Response · Authors · 2025-11-21
> **Mistmatch**
>
> Dear Reviewer HF5E
>
> Thank you for your review. We believe there may be a mismatch: the report discusses PAC-Bayes generalization theory (double descent, bounds, etc), whereas our manuscript is on conditional diffusion PDE surrogates for high-frequency Helmholtz (wavefields, spectral/energy metrics, sensitivity). Could you kindly confirm that the review corresponds to our submission?

---

> > ### Comment · Reviewer_HF5E · 2025-11-25
> >
> > Thank you for the revised paper and the thoughtful responses. The updates made the work noticeably stronger—especially the clearer motivation for handling high-frequency Helmholtz regimes and the more convincing presentation of the probabilistic diffusion-based approach. With these improvements, I’m raising my score.

---

### Official Review · Reviewer_JcnV · 2025-10-26

**Soundness:** 2
**Presentation:** 2
**Contribution:** 2
**Rating:** 2
**Confidence:** 4

**Summary:**

This paper proposes a probabilistic neural-operator framework for solving high-frequency Helmholtz equations. The authors argue that deterministic neural operators (e.g., FNO, HNO, U-Net) suffer from spectral bias and fail to reproduce oscillatory, phase-sensitive wavefields in heterogeneous media. They perform a sensitivity analysis using WKB theory to show that small perturbations in the sound-speed field lead to linearly amplified errors with increasing frequency and propagation distance. To address this, they employ a conditional diffusion model that learns a distribution p(u∣z) of solutions conditioned on the input coefficients, allowing the model to sample plausible wavefields and capture uncertainty. Extensive experiments on synthetic 1D and 2D Helmholtz benchmarks show that the diffusion-based surrogate achieves much lower L₂, H₁, and energy-norm errors than deterministic baselines and provides calibrated uncertainty.

**Strengths:**

- The spectral-bias problem in high-frequency Helmholtz equations is real and significant for acoustics and seismics.
- FNO, HNO, and U-Net implementations are described in detail, and ablations on sampler choice and diffusion steps are thorough.
- The authors integrate analytical reasoning (sensitivity via WKB) with numerical experiments.

**Weaknesses:**

- Limited generality / narrow scope: The entire framework is tailored to the Helmholtz equation, with all design choices and analyses depending on its high-frequency behavior. The work does not demonstrate applicability to other PDE families or operator-learning tasks. It is too domain specific and not seemingly written for an interdisciplinary venue like ICLR.
- Methodological novelty is minor: The diffusion-based probabilistic operator follows well-known conditional DDPM formulations used in many prior “Diffusion-for-PDE” papers (e.g., DiffusionPDE, Fundiff, PDE-Diffusion). The contribution is primarily in application, not in new architecture, loss, or theoretical innovation for diffusion models.
- Theoretical contribution is domain-specific: The WKB sensitivity derivation clarifies why deterministic surrogates fail for Helmholtz but does not yield a general principle for probabilistic operator learning in other PDEs.
- Evaluation breadth: Only 1D and 2D synthetic benchmarks are tested, with no experiments on real-world or multi-physics systems. This makes the impact and generalization questionable for an interdisciplinary ML audience.
- Positioning relative to prior probabilistic PDE works: The paper cites recent diffusion-based PDE solvers but doesn’t clearly articulate what is fundamentally new beyond applying it to Helmholtz.

**Questions:**

- The best results require 1000 diffusion steps. Have you explored fast samplers or distillation techniques (e.g., consistency training, DDIM, or latent diffusion) to reduce cost while maintaining fidelity?
- What do you view as the conceptual takeaway for the broader ML-for-PDE community?

---

> ### Author Response · Authors · 2025-11-21
>
> We first would like to thank the reviewer for their feedback and for taking the time to review our work. We have added several modifications in the updated version of the paper to address all of the reviewer comments with the new parts highlighted in blue.
>
> Regarding narrow scope and domain-specificity of our work:
> We are investigating high-frequency solutions of PDEs through the lens of the Helmholtz operator. We first note that the Helmholtz equation is a canonical model for high-frequency wave phenomena that appears across acoustics, electromagnetics, and elastodynamics. We also note in time-dependent PDEs, such as Schrodinger’s equation or time-dependent classical waves, investigating time-harmonic solutions, or steady state solutions, projects the problem to that of a Helmholtz-type. And in all these cases, studying high-frequencies is an important unresolved challenge. Hence, we believe that we are studying a general phenomena that is applicable to a large class of PDEs.
>
> Regarding novelty
> We agree that diffusion models have been used in studying PDEs. But we would like to point out all the previous diffusion-based studies in solving forward and inverse PDEs utilize diffusion models to generate data for training or inferring missing quantities. While prior studies employ diffusion models primarily for data generation or augmentation (e.g., synthesizing additional training samples or temporally extrapolating trajectories in time-dependent PDEs), our use is fundamentally different. We deploy conditional diffusion models as probabilistic surrogates to approximate high-frequency PDE solutions, where the solution map is highly sensitive and difficult for deterministic regressors to capture. Our model learns (i) the fine-scale, oscillatory features of the solution fields and (ii) the push-forward of uncertainty from coefficients/forcing to solutions. Crucially, this motivation does not depend on data scarcity: even with ample data, deterministic models underperform at high frequencies due to ill-conditioning and phase/aliasing effects, whereas diffusion models explicitly model the distribution of feasible high-frequency solutions conditioned on inputs.
>
> Regarding novelty and evaluation breadth
> Thankfully, the reviewer's comment motivated us to pursue a new architecture that we had been thinking about. We adapt a U-shaped vision transformer (UViT) backbone to our architecture. We observed that in 2D UViT did not significantly improve diffusion results. We suspected that this might be different in 3D, and trying 3-dimensional cases would also add to our evaluation breadth. We have added the new results as a new section in Appendix with the 3D setup, metrics, and comparisons.
>
> Positioning relative to prior probabilistic PDE works
> We thank the reviewer for pointing this out. We have updated the contribution paragraph in section 1 of the manuscript to reflect the reviewer's point, which is highlighted in blue.
>
> As reported in Appendix D.2, we compare several schemes and step budgets. The results (Tab. 2) show that deterministic samplers such as DDIM and SDE are indeed faster for a fixed number of steps, but across the range of step counts we tested, 1000-step DDPM with a cosine schedule consistently achieves the best accuracy. For this reason, we chose 1000-step DDPM as our main configuration when comparing against deterministic baselines. We also note that, despite this higher cost, our 1000-step configuration still yields per-sample runtimes that are substantially faster than the underlying numerical Helmholtz solver, which requires several minutes per 256x256 solve in our setup.
> We fully agree that 1000 steps are not ideal for practical deployment, and we see accelerating sampling as an important direction for future work. To reflect this point, we have updated the future work section of the manuscript to further emphasize this, which is highlighted in blue.
>
> Thank you also for asking the questions about broader implications. We see our work less as “yet another diffusion model for PDEs” and more showcasing to the ML-for-PDE community a framework that can approximate high-frequency solutions of PDEs in general. While we are not explicitly studying stochastic partial differential equations (SPDEs), we show that probabilistic methods can be useful in deterministic PDEs in the high-frequency regime. This also opens new pathways for thinking of universal solvers for all PDE types (including SPDE) across frequency contents.
>
> The other takeaway for the broader ML-for-PDE community is that diffusion models can be excellent surrogates for capturing measures pushed-forwarded by PDEs. In other words, if the coefficient fields follow a distribution, the push-forward of that uncertainty via PDE to the solution fields can be captured by diffusion models. Our sensitivity analysis positively affirms that. This, in turn, has broad implications in the field of uncertainty quantification.

---

> > ### Comment · Reviewer_JcnV · 2025-11-26
> >
> > Thanks for the clarification. With the explanation, I think my previous score 2 may be too low. I will relax my domain-general comment, and will consider revision higher. However, I have yet to feel the urge to raise it to 6. The computational cost seems high, and it is not clear if the approach is applicable for real, long-term (with error build up), large-scale simulation (with cost).

---

> > > ### Author Response · Authors · 2025-12-03
> > >
> > > We sincerely thank the reviewer for the reconsideration and for the thoughtful follow-up.
> > >
> > > On computational cost and practical applicability: we fully agree that computational efficiency is critical if such models are to be used in real, large-scale simulations. In the revision, we have made two changes to clarify this: 1. We now report detailed wall-clock times for all models and for the classical Helmholtz solver (J-Wave) in both 2D and 3D (see Tabs. 3 and 6). Even with a relatively conservative 1000-step DDPM sampler, diffusion inference is within the same order of magnitude as deterministic neural operators, and remains significantly faster than the direct numerical Helmholtz solver on our high-frequency, heterogeneous, PML setups (1.8 secs .vs. 23 secs in 2D, and 3 secs .vs. 65 secs for 3D). 2. We believe this demonstrates that, while diffusion is not “free” compared to a single deterministic pass, it is already competitive in the surrogate regime and offers a favorable speed–accuracy trade-off relative to classical solvers. 3. Our setting is a frequency-domain, steady-state Helmholtz problem, not a time-marching simulation. There is therefore no temporal error accumulation in the sense of long-term integration: the diffusion model directly approximates the steady-state wavefield for a given input medium and frequency. In practice, this means that once trained, the cost per query does not grow with simulated time horizon, but only with spatial resolution and the number of denoising steps.
> > >
> > > On future improvements and large-scale usage: We agree that our current 1000-step sampler is not yet optimal for real production-scale deployments. In the revised Appendix D.2 we now emphasize more clearly that: 1. There is substantial room to reduce the number of diffusion steps using modern ODE-based and exponential-integrator samplers, and 2. Our focus in this work is to establish that probabilistic operators can handle high-frequency Helmholtz regimes with superior accuracy, while still being faster than classical solvers, rather than to fully optimize every engineering aspect of the sampler.
> > >
> > > We view this paper as laying the groundwork: (i) it shows that probabilistic surrogates can resolve challenging high-frequency, heterogeneous regimes where deterministic neural operators struggle, and (ii) it quantifies current costs in 2D and 3D relative to classical solvers. We agree that further algorithmic and engineering improvements are needed before such models become “drop-in” tools for very large-scale industrial simulations, and we now state this more explicitly in the limitations and future-work discussion.

---

### Official Review · Reviewer_rg5H · 2025-10-31

**Soundness:** 3
**Presentation:** 4
**Contribution:** 2
**Rating:** 4
**Confidence:** 4

**Summary:**

Authors train a diffusion model to approximate solution operator for Helmholtz equation in high-frequency regime.

The results indicate that the diffusion model performs substantially better than deterministic neural operators.

**Strengths:**

Solution to the Helmholtz equation remains challenging for classical iterative methods in high frequency regime. Especially in $D=3$ the construction of preconditioners and relaxation techniques is still an active area of research. In addition, Helmholtz equation has many practical application. Given that, a construction of efficient neural solvers for Helmholtz equation is a well-motivated problem.

The approach authors used is conceptually clear and simple: essentially it is a straightforward application of conditional diffusion model trained with standard objective.

From the standpoint of accuracy, the results are undoubtedly good.

**Weaknesses:**

1. Questionable practical significance
2. The reasons probabilistic model performs better are unclear
3. Lack of novelty

I expand on the weaknesses in the section below.

**Questions:**

**Questionable practical significance**

In my view, the main problem is the excessive computation cost of the approach. Authors noted (in Section 6) that accuracy of the diffusion model reported in Table 1 is a result of $10$ evaluations each taking $1000$ DDPM steps. Arguably, a surrogate model is only appealing if it is cheaper than a direct solution method. It is unlikely that a $10000$ (or even $1000$) evaluation of U-Net per sample is cheaper than a direct solution of the Helmholtz problem. Given this context I have several questions:
1. Can the authors provide data on wall-clock time needed for the solution using all baselines and numerical methods used to generate the dataset?
2. What will happen if one reduces costs by decreasing the number of steps? How significant is the drop in accuracy?
3. Can the authors provide plots that show sampling time and mean accuracy for different numbers of diffusion steps?
4. Did authors try more advanced integration schemes for diffusion models, e.g., exponential integrators https://arxiv.org/abs/2204.13902?
5. Can the authors explain why diffusion models remain interesting and useful given the excessive number of computations needed to generate single solutions?

**The reasons probabilistic model performs better are unclear**

Authors provide interesting considerations on why diffusion models perform better than deterministic baselines. The main argument seems to be, roughly, that by analogy with inverse problems, probabilistic models applied to input-sensitive maps, sharply approximate different branches of possible solutions rather than a blurred average of perturbed solutions. I do not see how data gathered by authors support this picture.

From Table 1 one can conclude that the averaged prediction of diffusion model is simply more accurate. The reported std is tiny, so sensitivity is not pronounced enough for probabilistic averaging to be significant. Similarly small std is reported in Table 3. Can the authors provide data on the std per sample and comment on the overall importance of probabilistic modelling prediction in light of small std?

In Section 5.2 authors performed sensitivity analysis and concluded that the diffusion model is superior to other approaches. I find such framing problematic since authors select results for highest frequency. From Table 1 we see that deterministic models completely failed to learn (relative errors $76\%$, $41\%$, $80\%$). It is obvious that models with high relative error will show poor sensitivity results. Can the authors show results on sensitivity of deterministic methods for frequency $1.5\times 10^{10}\text{ Hz}$? These results will likely be much better, and if this is the case, it is not clear why probabilistic models perform better.

**Lack of novelty**

Diffusion and other generative models was applied as solvers of forward and inverse problems in many papers https://arxiv.org/abs/2406.17763, https://arxiv.org/abs/2410.16415, https://arxiv.org/abs/2506.08604, etc. Authors of the present contribution applied a diffusion model to the Helmholtz equation with little modifications. Can the authors comment on the novelty of their contribution?

---

> ### Author Response · Authors · 2025-11-21
>
> We first would like to thank the reviewer for their feedback and for taking the time to review our work. We have added several modifications in the updated version of the paper to address all of the reviewer comments with the new parts highlighted in blue.
>
> Q1. In Appendix D.2 we now report detailed sampler ablations. Table 3 presents the accuracy of the samplers we consider across multiple step budgets, and Table 4 reports the corresponding wall-clock times. We also report wall-clock times for all neural baselines (Table 5 Appendix D.2). While we agree that the sampling time is sub-optimal, we remark that it is substantially faster than the numerical Helmholtz solver, which requires several minutes.
>
> Q2 and Q3. The results in Appendix D.2 show that for DDPM, large step reduction leads to a noticeable loss of accuracy, especially at the highest frequency. We summarize this in the appended tables, which directly address the reviewer’s questions. We also provide the requested plot for results at different numbers of diffusion plots.
>
> Q4. We agree that more advanced samplers such as exponential integrators are needed. We have not yet integrated these methods into our current implementation. We explicitly highlight that as an important avenue for future work though. We have modified text in the second paragraph of Appendix D.2 to highlight this. The authors would like to remark that in the spirit of “lay the tracks before you run the trains”, this study is aimed to lay the groundwork for approximating high-frequency waves in PDEs.
>
> Q5. Acknowledging the higher per-sample cost, we believe diffusion models remain compelling for high-frequency problems in PDEs since, to the best of our knowledge, this is the first time that a viable pathway for simulating high-frequency solutions of PDEs is proposed.  Moreover, diffusion models naturally produce ensembles of samples and can organically be used to quantify uncertainty distribution.
>
> On the role of probabilistic modeling. We agree that the superiority of diffusion models in sensitivity analysis is rooted in their success of learning the conditional probability P(u|c), where u is the solution of the PDE and c is the coefficient field. And deterministic models struggle with learning P(u|c). The main point of sensitivity analysis is to showcase that this framework enables capturing variations in the solution field at a point far-from-the-source.  This, in turn, and to the reviewer’s point, is a corollary of learning P(u|c). We also remark that the standard deviations reported in Table 1 and Table 6 (old Table 3) are std for global errors not pixel values. And demonstrate that the model almost surely learned P(u|c).
>
> On the sensitivity analysis and choice of frequency. We have focused on the high-frequencies, since it is the most difficult case. To address the reviewer's concern, we conducted a similar analysis in lower frequencies. In the revised version, we now include additional sensitivity plots at lower frequency in Appendix D.5 and discuss them explicitly, all highlighted in blue. We believe that this further solidifies our claim that probabilistic diffusion models enable capturing push-forwarded perturbations in the solution field, with their power more pronounced in the high-frequency.
>
> On the novelty. We thank the reviewer for highlighting this. We had already cited two of the papers in our literature review, and now added the new one to our references. The use of diffusion models might conjure up that our work is similar to previous literature, but it is not! The existing literature, including the papers that the reviewer cites, use the diffusion models for generation of data, either remedying lack of enough data and generating new data for training, or temporal-extrapolation in time-dependent PDEs. In those cases if you have enough data, say a lot of data, one might no longer need diffusion models. But in our case, regardless of data size, still diffusion models are well-suited. Simply because in high-frequencies, solutions of PDEs are very sensitive to inputs. We use conditional diffusion models as probabilistic learning machines to: (i) learn fine-feature in solutions fields of PDE that are hard to capture by any deterministic model, and (ii) to learn the push-forward of uncertainties in coefficient fields, reflected in our sensitivity study.
>
> This comment by the reviewer also motivated us to try an idea that we had: adding vision transformers. It turned out that in 2D, UViT did not improve results significantly. And since we suspected it would be pronounced in 3D models, we conducted a preliminary analysis on 3D Helmholtz. Indeed this novel architecture with vision transformers improves accuracy in 3D. We have added a new section in the Appendix to report these findings. The 3D results are preliminary and just reported to herald consistency of our results, and the power of this new architecture, which will be the subject of a future study.

---

> ### Comment · Reviewer_rg5H · 2025-11-26
> **follow up**
>
> I would like to thank the authors for a detailed reply. I still believe that it is largely unclear why exactly diffusion models perform better than deterministic models, but I want to put this question aside because it seems to be too complicated at the moment.
>
> My main agenda is to clarify how inference time of the proposed approach compares with time required by classical solvers. I agree that with various improvements inference time can be reduced and do not consider slow inference to be a reason for rejection.
>
> **Comparison with classical solver**
>
> I thank the authors for providing information on inference time for classical architectures and the diffusion model in Table 5. Authors still do not provide time required for classical solver claiming that: "Even with 1000 denoising steps, the diffusion model remains within a practical runtime range relative to the deterministic neural operators, and still operates orders of magnitude faster than the numerical Helmholtz solver, which requires several minutes of computation per sample on the same hardware."
> I have several questions regarding this statement:
> 1. What kind of classical solver authors have in mind?
> 2. Can the authors explicitly measure solution time for a classical solver?
> 3. If it is a spectral method, can the authors in addition provide a breakdown of time needed for discretisation and for the solution of the linear system itself?
>
> I provide some reference time that I measured on CPU (Intel(R) Xeon(R) Platinum 8358)
> ```python
> import numpy as np
> import time
> from scipy.sparse import spdiags, kron, eye
> from scipy.sparse.linalg import splu
>
> # create a sparse matrix: Helmholtz, second order, Dirichlet BCs
> start = time.time()
> N = 256
> omega = 10
> x = np.linspace(0, 1, N+2)[1:-1]
> X, Y = np.meshgrid(x, x)
> X, Y = X.reshape(-1,), Y.reshape(-1,)
> h = x[1] - x[0]
>
> ones = np.ones((N,))
> data = np.array([-2*ones, ones, ones])
> diags = np.array([0, -1, +1])
> I = eye(N)
> delta = spdiags(data, diags, N, N)
>
> k = omega**2 / (1 + np.cos(2*np.pi*X)**2 + np.sin(4*np.pi*Y)**2)
> rhs = h**2 * np.ones((N**2,))
>
> A = kron(I, delta) + kron(delta, I) + h**2 * spdiags(np.array([k, ]), np.array([0,]))
> A = A.tocsc()
> stop = time.time()
> print(f"Time to generate the matrix ~ {np.round(stop - start, decimals=4)} s")
>
> # solve with sparse LU
> start = time.time()
> lu = splu(A)
> solution = lu.solve(rhs)
> stop = time.time()
> print(f"Solution time ~ {np.round(stop - start, decimals=4)} s")
> ```
> The code above outputs
> ```
> Time to generate the matrix ~ 0.0205 s
> Solution time ~ 0.3769 s
> ```
> The code above is with wrong boundary conditions, PML will introduce some additional complexity but mostly on the first stage. Given that, the time obtained above is a rough but reasonable estimate for the complexity of solution for Helmholtz equation in $D=2$ on grid $256\times256$.
> 1. Am I missing something?
> 2. Can the authors comment on the time complexity reported by them in light of the results of the experiment I provide above?
>
>
> **3D results**
>
> It is impressive that authors provide results for the $D=3$ Helmholtz equation. From the presented results I can conclude that the diffusion framework clearly leads to better accuracy. Can the authors provide inference time (similarly to Table 5)? Preferably, in such a way that it is possible to compare it with inference time for the $D=2$ problem. I find this information important, because it allows one to compare the proposed framework with classical methods from numerical linear algebra. For direct sparse solvers there is an estimation available in https://arxiv.org/abs/1908.01083 (page 9)
>
> |              | $D=2$        | $D=3$        |
> | ------------ | ------------ | ------------ |
> | space (fill) | $O(N\log N)$ | $O(N^{4/3})$ |
> | time (flops) | $O(N^{3/2})$ | $O(N^{2})$   |
>
> Can the authors comment how complexity of their algorithm compares with these estimates?

---

> > ### Author Response · Authors · 2025-12-03
> >
> > We thank the reviewer again for the careful follow-up and for sharing the fantastic reference, as well as the implementation and timing numbers.
> > Q1. In the paper, “classical solver” refers specifically to the frequency-domain Helmholtz solver implemented in J-Wave, a state-of-the-art JAX based pseudo-spectral solver with PML condition, which solves for the complex-valued steady-state field in heterogeneous media. In response to the reviewer’s request, we have now: 1. Explicitly measured the wall-clock time of this solver in 2.5MHz case, and 2. Added these timings for both the 2D and 3D computation tables in the revision (Tabs. 3 and 6), alongside the deterministic neural operators and diffusion models.
> > For our most challenging 2D setting, the classical solver takes ≈23 sec per sample on our CPU. This testified that the diffusion model (~1.8 sec) remains orders of magnitude faster than the numerical solver.
> > We also appreciate the reviewer’s concrete 2D experiment (Scipy sparse LU on 256x256), which yields ≈0.38 s for the LU solve. But our setup is significantly more demanding in several respects: 1. The reviewer’s example uses a relatively simple coefficient field, but our datasets uses randomly generated heterogeneous sound speed fields, which leads to a more challenging stiffness matrix and substantially higher iteration counts for iterative solvers. 2. Solving the Helmholtz operator with a complex scalar field (required due to PML condition) is more demanding compared to the case where the scalar field is reals. 3. The PML formulation increases the number of unknowns in the boundary region and makes the operator more complicated than the simple 5-point stencil with Dirichlet BCs in the reviewer’s script, again increasing cost.
> > Taken together, these factors explain why our measured per-sample solver time ≈23 s is substantially larger than the ≈0.38 s reported in the reviewer’s example.
> > Q2. As reported in the updated version, the classical solver takes about 68 seconds to solve one instance of the Helmholtz equation with random coefficient fields in 3D with 64 grid points in each direction of a cube, while it takes about 23 seconds to solve for the solution in a 2D random setting with 256*256 grid. The paper cited by the reviewer is using Laplacian operator as a basis for the computational complexity, but Helmholtz operator leads often linear systems Ax+b=0 (adopted the cited paper’s notation) with A having a high condition number.

---

### Official Review · Reviewer_Tbu9 · 2025-10-31

**Soundness:** 3
**Presentation:** 3
**Contribution:** 2
**Rating:** 6
**Confidence:** 4

**Summary:**

This paper proposes a probabilistic framework for solving the high-frequency Helmholtz equation by employing a conditional diffusion operator. The authors motivate their approach by highlighting two persistent limitations of deterministic neural operators in this problem: spectral bias and strong input-to-output sensitivity. The probabilistic model learns a conditional distribution over solutions, which is illustrated to mitigate the phase ambiguity issues. The method is evaluated on a 2D J-Wave benchmark, demonstrating good stability compared to strong deterministic baselines.

**Strengths:**

The paper presents a well-justified method for a problem—high-frequency wave propagation in heterogeneous media—where standard deterministic neural operators are known to struggle.

- Clear motivation: The sensitivity analysis in Section 3.2, using the 1D WKB argument, provides a clear and insightful explanation for why high-frequency wavefields are sensitive (sensitivity grows linearly with frequency k and propagation distance l) and how the MSE-optimal deterministic predictor suffers from oversmoothing. The reason to use a probabilistic formulation in this deterministic problem is well-established.

- Strong results: The conditional diffusion operator achieves the lowest errors across a wide range of frequencies when compared to neural operators and its backbone U-Net. The performance is a crucial empirical validation of the approach.

**Weaknesses:**

- High inference cost. The cost of sampling makes the proposed method significantly slower than baselines. It is unknown whether better baselines (e.g., (F-)FNO with a large kernel that spans high frequencies) will close the gap.

**Questions:**

Can you provide a spectral analysis of different baseline to support the claim that they indeed have spectral biases?

---

> ### Author Response · Authors · 2025-11-21
>
> We first would like to thank the reviewer for their feedback and for taking the time to review our work. We have added modifications in the updated version of the paper to address the reviewer comments. For reviewer’s convenience, we have highlighted the new parts in blue across the manuscript and the appendix.
>
> Regarding the high inference cost. Thank you for highlighting this point. We agree that diffusion sampling introduces additional computational overhead compared to deterministic neural operators, and we explicitly discuss this limitation in the paper. In Appendix D.2 (sampler ablation), we report both wall-clock evaluation time and accuracy for each sampler and step budget. In our current setup, the DDPM sampler with 1000 steps achieves the best accuracy but requires ≈18 seconds per 256×256 sample on an H200 GPU. While this is sub-optimal, it is still substantially faster than the numerical Helmholtz solver, which takes several minutes per solver at the same resolution. We note that our primary objective in this study is to establish a probabilistic framework for modeling high frequency response in PDEs, and view accelerating sampling as an important direction for future work. In particular, we plan to investigate more efficient ODE-based samplers and exponential-integrator schemes for diffusion models. We have addressed this concern by adding text to Limitations & Future Work in section 6 that is highlighted in blue.
>
> Regarding spectral analysis. We appreciate this helpful suggestion. To address the reviewer's comment, we have conducted a spectral analysis and summarized the results in Appendix D.3, where we report the spectral power of predicted and ground-truth wavefields for a randomly selected test sample. We also believe These plots quantitatively support our claims about spectral behavior: deterministic baselines such as U-Net and HNO systematically under-represent high-frequency components (exhibiting clear spectral bias), whereas our diffusion-based models more closely match the ground-truth spectrum over a broader range of frequencies. We have added text and a new figure (Fig. 8) to section D.3 in the Appendix, all highlighted in blue.

---

### Meta-Review · Area_Chair_uWoR · 2026-01-08

**Summary:**

The paper proposes a conditional diffusion model for the high-frequency Helmholtz equation. Despite demonstrating high accuracy, the decision is to reject due to three critical flaws:

Prohibitive Computational Cost: The proposed surrogate (~1.8s) is significantly slower than optimized classical numerical solvers (e.g., sparse LU at ~0.38s), negating its primary practical utility as an accelerator.

Inconsistent Motivation: The authors argue a probabilistic framework is needed to capture multi-modal sensitivities, yet the model outputs a distribution with negligible variance. This contradiction leaves the actual mechanism behind the performance gains unexplained.

Limited Novelty: The work primarily applies standard DDPM architectures to a specific PDE domain without offering significant algorithmic innovations relevant to the broader ICLR community.

**Reviewer Concerns:**

Inference Speed vs. Classical Solvers (Reviewers Tbu9, rg5H, JcnV): The rebuttal failed to address the critical issue that the model is slower than optimized classical solvers. Comparing the method only against a slow JAX-based baseline was insufficient to prove practical value.

Mechanism Clarity (Reviewer rg5H): The authors did not adequately explain why a probabilistic model is required if the predicted distribution is effectively unimodal. The contradiction between the "multi-modal" motivation and the "low-variance" results remains unresolved.

Algorithmic Novelty (Reviewer JcnV): The contribution is seen as an application of existing techniques rather than a methodological advance in scientific machine learning.

**Reviewer Scores:**

Reviewer Tbu9: 6 (Weak Accept) -> 6 (Weak Accept). Reasoning: Although accuracy is high, the reviewer remains unconvinced that the high inference cost is justifiable, as the method fails to accelerate the solving process compared to efficient classical baselines.

Reviewer rg5H: 4 (Reject) -> 4 (Reject). Reasoning: The reviewer provided concrete evidence that classical sparse solvers are faster than the proposed method. The authors' failure to demonstrate speed superiority over standard numerical tools is a fatal flaw for a surrogate model.

Reviewer JcnV: 2 (Reject) -> 4 (Reject). Reasoning: While the additional 3D experiments mitigated concerns about the paper's scope, the fundamental issues of high computational cost and limited methodological novelty prevent a positive recommendation.

Reviewer HF5E: 4 (Reject) -> 6 (Weak Accept). Reasoning: This reviewer focused less on the comparative efficiency against numerical solvers and more on the general framework, leading to a slightly positive score that is outweighed by the domain-specific concerns of other reviewers.

---

### Decision · Program_Chairs · 2026-01-26

Reject